# Structural mechanism of human oncochannel TRPV6 inhibition by the natural phytoestrogen genistein

Arthur Neuberger [1], Yury A. Trofimov [2], Maria V. Yelshanskaya[1], Kirill D. Nadezhdin [1], Nikolay A. Krylov [2], Roman G. Efremov[2] & Alexander I. Sobolevsky [1] ✉

Calcium-selective oncochannel TRPV6 is the major driver of cell proliferation in human cancers. While significant effort has been invested in the development of synthetic TRPV6 inhibitors, natural channel blockers have been largely neglected. Here we report the structure of human TRPV6 in complex with the plant-derived phytoestrogen genistein, extracted from *Styphnolobium japonicum*, that was shown to inhibit cell invasion and metastasis in cancer clinical trials. Despite the pharmacological value, the molecular mechanism of TRPV6 inhibition by genistein has remained enigmatic. We use cryo-EM combined with electrophysiology, calcium imaging, mutagenesis, and molecular dynamics simulations to show that genistein binds in the intracellular half of the TRPV6 pore and acts as an ion channel blocker and gating modifier. Genistein binding to the open channel causes pore closure and a two-fold symmetrical conformational rearrangement in the S4−S5 and S6-TRP helix regions. The unprecedented mechanism of TRPV6 inhibition by genistein uncovers new possibilities in structure-based drug design.

TRPV6 is a representative of the vanilloid subfamily of transient receptor potential (TRP) channels that serves as an entry gate for capturing dietary calcium ions in the gut[1–3]. TRPV6 mutations and abnormal expression of this channel[4–8] have been linked to a range of human diseases associated with disturbed calcium homeostasis, including transient neonatal hyperparathyroidism, undermineralization, and dysplasia of the human skeleton, hypercalciuria, chronic pancreatitis, various reproductive diseases, Pendred syndrome and Crohn's-like disease[4,9–20]. Since calcium uptake is linked to cell proliferation and cancer progression, TRPV6 was also declared an oncochannel[21–23]. Indeed, TRPV6 was found to overexpress in some of the most severe human cancers, including leukemia, breast, prostate, colon, ovarian, thyroid, and endometrial cancers[5–7,21,24]. In addition, an ancestral variant of this oncochannel has emerged as a driver of higher incidence, higher mortality, and more aggressive forms of different

cancer types in people of African descent[25]. Inhibitors of TRPV6 are therefore in high demand.

While several synthetic inhibitors of TRPV6 have been making a slow progress towards clinical trials[1–3,26–28], scientific exploration and pharmaceutical exploitation of natural TRPV6 inhibitors have been largely neglected despite their apparent benefits, such as pharmacokinetics optimized by nature in the course of evolution[29]. The natural isoflavone and phytoestrogen genistein (4′,5,7-trihydroxyisoflavone) was previously suggested to act as a TRPV6 inhibitor[30]. Genistein is a common precursor in the biosynthesis of antimicrobial phytoalexins and phytoanticipins in legumes and as a predominant isoflavone in nutritional soy products it has a potential to be the major component of an individual's diet[31–34]. Importantly, dietary genistein shows a range of potential health-beneficial effects, including the ability to inhibit cell invasion and metastasis in various forms of human cancer[31,32,35–43].

[1]Department of Biochemistry and Molecular Biophysics, Columbia University, New York, NY, USA. [2]Shemyakin-Ovchinnikov Institute of Bioorganic Chemistry, Russian Academy of Sciences, Moscow, Russia. ✉e-mail: as4005@cumc.columbia.edu

Beyond treatment of prostate, colon, kidney, pancreatic, ovarian, breast, and lung cancers[38,44–67], therapeutic potential of genistein extends to the treatment of cardiovascular diseases[68–70], post-menopausal[71,72] and gastrointestinal[73] ailments, and bone loss[74–77]. Genistein has been investigated in 75 clinical trials (clinicaltrials.gov), where it shows antimetastatic efficacy[78] and positive effects in treatment of metabolic syndrome[79].

In this study, we explore the molecular basis of human TRPV6 (hTRPV6) inhibition by the natural phytoestrogen genistein[30] extracted from *Styphnolobium japonicum*. Using cryo-electron microscopy (cryo-EM) combined with calcium imaging, electrophysiology, mutagenesis, and molecular dynamics (MD) simulations we show that genistein binds in the intracellular half of the hTRPV6 pore and acts as an ion channel blocker and gating modifier. Upon binding to the open pore of hTRPV6, genistein converts it into a non-conducting conformation with a two-fold symmetrical arrangement of the pore-forming segments. This conformation is likely stabilized by two putative metal binding sites at the pore intracellular entry and has S4–S5 and S6-TRP regions, which typically include two α-helices, transformed into three helices-containing segments in two diagonal subunits. The unusual mechanism of TRPV6 inhibition by genistein lays the foundations for the development of much-needed new drugs targeting TRPV6-associated diseases, including cancers.

## Results

### Functional characterization of hTRPV6 inhibition by genistein
TRPV6 is a constitutively open ion channel[80–82]. In response to a –100 to +70 mV voltage ramp, whole-cell patch-clamp recordings from HEK 293S cells expressing wild-type human hTRPV6 showed inward-rectified currents (Fig. 1a), typical for this ion channel[83–85]. In the presence of genistein, the TRPV6-mediated currents were reduced. Thus, at –60 mV membrane potential, 50 μM genistein produced $54 \pm 4\%$ (mean ± SEM, $n = 13$) inhibition of the hTRPV6-mediated current. Measurements of the concentration dependence of the hTRPV6-mediated current inhibition yielded the half-maximal inhibitory concentration, $IC_{50} = 40.7 \pm 2.6$ μM, and the Hill coefficient, $n_{Hill} = 1.80 \pm 0.13$ ($n = 7$, Supplementary Fig. 1). We also monitored hTRPV6 inhibition by genistein using Fura-2-based measurements of changes in intracellular $Ca^{2+}$. Changes in the fluorescence intensity ratio at the excitation wavelengths 340 and 380 nm ($F_{340}/F_{380}$) evoked by addition of 10 mM $Ca^{2+}$ were measured after pre-incubation of hTRPV6-expressing HEK 293S cells in different concentrations of genistein (Fig. 1b). Genistein inhibited hTRPV6-mediated $Ca^{2+}$ uptake with the values of $IC_{50} = 113.2 \pm 4.7$ μM and $n_{Hill} = 0.77 \pm 0.02$ ($n = 3$, Fig. 1c).

### Cryo-EM analysis of hTRPV6 in the presence of genistein
Purified hTRPV6 protein was supplemented with 2 mM genistein and subjected to cryo-EM analysis (Methods; Supplementary Fig. 2). Collected cryo-EM micrographs showed evenly dispersed particles of hTRPV6 (Supplementary Fig. 3a). Processing of the data revealed two distinct populations of particles, both showing diverse angular coverage. The corresponding 2D-class averages demonstrated clearly visible secondary structure elements, supporting the high quality of the collected cryo-EM data (Supplementary Fig. 3b, c). The first, smaller population of particles yielded a 2.71-Å cryo-EM map that showed four-fold rotational symmetry (C4) and represented a typical genistein-free apo state, hTRPV6_Open (Fig. 1d–f, Supplementary Fig. 3d, f, h and Supplementary Table 1). The second, predominant population of particles yielded a 2.66-Å 3D reconstruction with no symmetry imposed (C1), which had two densities of the size of a genistein molecule that had never been seen in TRPV6 reconstructions before (Fig. 1g–i, Supplementary Figs. 3e, g, i and 4, Supplementary Table 1). Further analysis showed that this second reconstruction represents a genistein-bound inhibited state of the channel, hTRPV6_GEN.

For the 4-fold symmetrical hTRPV6_Open homotetramer, we built a molecular model of a single subunit, including residues 28–637 and excluding residues 1–27 (N-terminus) and 638–725 (C-terminus), which were not clearly resolved in the cryo-EM map. We also built a molecular model for each of the four subunits of the hTRPV6_GEN homotetramer, including residues 27–638 and excluding residues 1–26 (N-terminus) and 639–725 (C-terminus) that were not clearly resolved in the corresponding cryo-EM density. The transmembrane regions of hTRPV6_Open and hTRPV6_GEN were surrounded by numerous non-protein auxiliary densities, which we modeled as annular lipids (Fig. 1d–i). Due to high quality of hTRPV6_Open and hTRPV6_GEN reconstructions, three such densities per subunit were identified as representing cholesteryl hemisuccinate (CHS), which was added to the protein sample during purification to enhance protein stability (see Methods). The CHS sites are likely to bind cholesterol in vivo.

### hTRPV6_Open and hTRPV6_GEN structures
The structures of hTRPV6_Open and hTRPV6_GEN (Fig. 2a–c) have a similar overall architecture to the previously determined structures of TRPV6[26,27,86,87]. In a nutshell, hTRPV6 is assembled of four subunits and contains a transmembrane domain (TMD) with a central ion channel pore and an intracellular skirt that is mostly built of ankyrin repeat domains connected by the three-stranded β-sheets, N-terminal helices, and C-terminal hooks[86]. Amphipathic TRP helices run nearly parallel to the membrane and interact with both the TMD and the skirt. The TMD is composed of six transmembrane helices S1–S6 and a re-entrant pore loop (P-loop) between S5 and S6. A bundle of the first four transmembrane helices represents the S1–S4 domain, which in voltage-gated ion channels forms a voltage sensor[88]. The pore domain of each subunit includes S5, P-loop, and S6, and is packed against the S1–S4 domain of the neighboring subunit in a domain-swapped arrangement[87,89].

Despite the overall similarity (Fig. 2c), hTRPV6_Open and hTRPV6_GEN structures have different conformations of the ion-conducting pore (Fig. 2d, e). As in all representatives of the tetrameric ion channel family, the ion-conducting pore of TRPV6 has two narrow regions, the selectivity filter formed by the extended regions of the P-loop and the gate formed by the S6 bundle crossing, separated by the central cavity in the middle. The selectivity filter of TRPV6 is a structural element responsible for high calcium selectivity and in both structures, it adapts a conformation nearly identical to the previously published TRPV6 structures. Interestingly, however, the gate region shows drastically different conformations. In hTRPV6_Open, the pore is wide open (Fig. 2f) and its narrow region is lined by residues N572 and I575 (Fig. 2d). This conformation is the same as in the previously published open-state structures of TRPV6, independent of whether they had C-terminus truncated or not and whether the protein was purified in different types of detergents, nanodiscs or amphipols (Supplementary Fig. 5), strongly supporting our initial assignment of hTRPV6_Open to the open conducting state.

In contrast to hTRPV6_Open, the gate region in hTRPV6_GEN is narrow and fully sealed by the side chains of L574 and M578 (Fig. 2e, f). Similar pore narrowing was observed in the closed-state structures of TRPV6, independent of whether they were apo-state structures[26,27,87] or structures obtained in the presence of inhibitors 2-APB or econazole, or ion channel blocker ruthenium red (RR)[27,90] (Fig. 2f). The characteristic feature of the TRPV6 closed-state is α-helical S6 (Fig. 2e). In the open state, S6 undergoes an α-to-π transition, which produces a π-bulge in the middle of this helix (Fig. 2d). Formation of the π-bulge is accompanied by a -100° rotation of the C-terminal portion of S6, which brings a completely different set of residues to face the pore and opens it for ion conduction[87,90].

### Two-fold symmetry of hTRPV6_GEN structure
A closer look at the hTRPV6_GEN structure reveals non-equivalency of the two pairs of diagonal subunits, A/C and B/D (Fig. 3). Overall, the

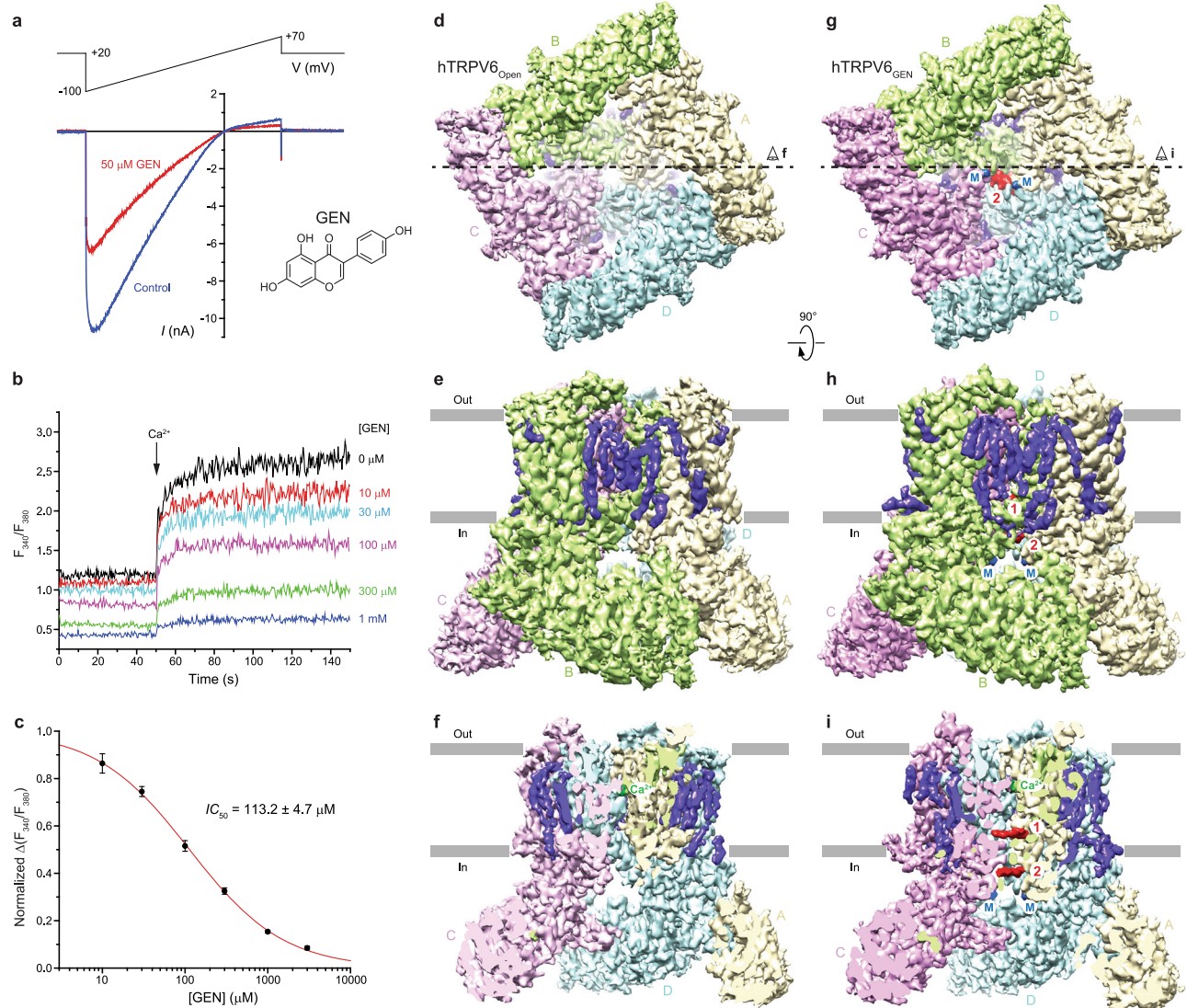

**Fig. 1 | Functional and cryo-EM characterization of TRPV6 inhibition by genistein. a** Whole-cell patch-clamp currents recorded from HEK 293S cell expressing hTRPV6 in response to −100 to +70 mV voltage ramp in the absence (blue) and presence (red) of 50 μM genistein. The inset shows the chemical structure of genistein. **b** Representative ratiometric Fura-2-based fluorescence measurements of changes in intracellular $Ca^{2+}$ for HEK 293S GnTI⁻ cells expressing hTRPV6. The changes in the fluorescence intensity ratio at 340 and 380 nm ($F_{340}/F_{380}$) were monitored in response to application of 10 mM $Ca^{2+}$ (arrow) after pre-incubation of cells with various concentrations of genistein. The experiment was repeated independently three times with similar results. **c** Dose-response curve for genistein inhibition of $Ca^{2+}$ uptake. The changes in the fluorescence intensity ratio at 340 and 380 nm ($F_{340}/F_{380}$) evoked by the addition of 10 mM $Ca^{2+}$ after pre-incubation with various concentrations of genistein were normalized to the maximal change in $F_{340}/F_{380}$ after the addition of 10 mM $Ca^{2+}$ in the absence of genistein. The curve through the data points is a fit with the logistic equation, with the mean ± SEM values of the half-maximum inhibitory concentration ($IC_{50}$), 113.2 ± 4.7 μM, and the Hill coefficient, $n_{Hill} = 0.77 ± 0.02$ ($n = 3$ independent experiments). **d–i** 3D cryo-EM density for TRPV6$_{Open}$ (**d–f**) and TRPV6$_{GEN}$ (**g–i**) viewed from the bottom (**d, g**), side (**e, h**) or side but cut off along the dashed lines in **d** and **g** (**f, i**), respectively. TRPV6 subunits are colored yellow, green, pink, and cyan. Putative densities for genistein at sites 1 and 2, lipids, and ions (M) are shown in red, purple, and blue, respectively. The density for calcium at the selectivity filter is shown in green.

structure is ~4-fold symmetrical but shows noticeable deviations from the 4-fold symmetry at the intracellular region of the ion channel pore, around the two putative sites of genistein binding (Fig. 3a, b). The exact regions that underlie the deviation from the 4-fold symmetry can be easily pinpointed by superposition of individual hTRPV6$_{GEN}$ subunits (Fig. 3c) and include the S4–S5 linker (Fig. 3d) and S6-TRP helix connection (Fig. 3f). Indeed, the conformations of the S4–S5 linker and S6-TRP helix connection appear to be similar within A/C and B/D pairs of the diagonal subunits and different between these two pairs (Fig. 3c and Supplementary Movie 1).

In the A/C pair, the S4–S5 region includes a continuous helical segment with two sections, S4–S5 linker and S5, tilted with respect to each other by ~28° at around W495 (Fig. 3d). This continuous helical

segment is present in all previously published 4-fold symmetrical structures of TRPV6 but the tilt angle, ~51°, is much larger than in A and C subunits of hTRPV6$_{GEN}$ (Fig. 3e). As a consequence of the smaller angular tilt, the unfolded region connecting S4 to the S4–S5 linker in A and C subunits of hTRPV6$_{GEN}$ (Fig. 3d) is much larger than the corresponding region in the 4-fold symmetrical structures of TRPV6 (Fig. 3e). The conformation of the S4–S5 region in subunits B and D of hTRPV6$_{GEN}$ shows an even more drastic difference from the 4-fold symmetrical structures. In this case, S5 is straight, elongated, and separated from the helical part of the S4–S5 linker by an unfolded stretch of polypeptide (Fig. 3d). Correspondingly, the S4–S5 region, which includes only two continuous helical segments in all published structures of TRPV6 (Fig. 3e), in B and D subunits of hTRPV6$_{GEN}$

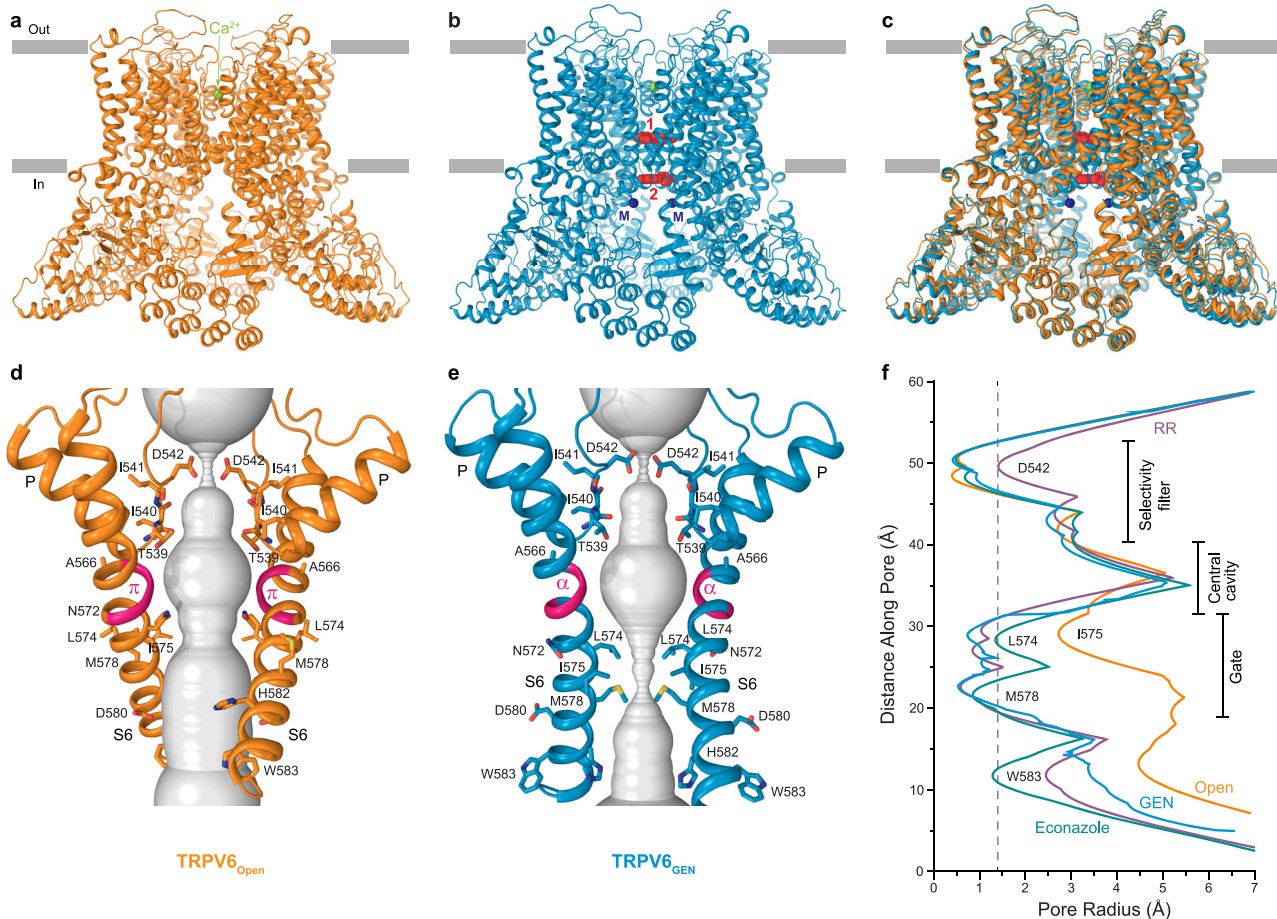

**Fig. 2 | Open-state and genistein-bound closed-state structures of TRPV6.**
**a**, **b** TRPV6$_{Open}$ (**a**) and TRPV6$_{GEN}$ (**b**) structures viewed from the side, with calcium at the selectivity filter shown as green spheres. In TRPV6$_{GEN}$, molecules of genistein at sites 1 and 2 are shown as red space-filling models, and metal ions at the intracellular pore entry (M) as dark blue spheres. **c** Superposition of TRPV6$_{open}$ (orange) and TRPV6$_{GEN}$ (blue) structures. **d**, **e** Pore-forming domain in TRPV6$_{open}$ (**d**) and TRPV6$_{GEN}$ (**e**) with the residues contributing to pore lining shown as sticks. Only two of four subunits are shown, with the front and back subunits omitted for clarity. The pore profile is shown as a space-filling model (gray). The region that undergoes the α-to-π transition in S6 is highlighted in pink. **f** Pore radius for TRPV6$_{Open}$ (orange), TRPV6$_{GEN}$ (blue), ruthenium red-bound structure TRPV6$_{RR}$ (purple, PDB ID: 7S8B), and econazole-bound structure TRPV6$_{Eco}$ (green, PDB ID: 7S8C), calculated using HOLE. The vertical dashed line denotes the radius of a water molecule, 1.4 Å.

transforms into the region with three helical segments (Fig. 3d and Supplementary Movie 1).

The S6-TRP helix region in subunits A and C of hTRPV6$_{GEN}$ includes two continuous helical segments (Fig. 3f), similar to the corresponding region in the previously published 4-fold symmetrical structures of TRPV6 (Fig. 3g). However, the α-helical S6 in the closed-state structures is one-helical turn shorter than S6 in subunits A and C of hTRPV6$_{GEN}$, while the similarly long S6 in the previously published open-state structures contains the π-bulge in the middle. The elongation of S6 in subunits A and C of the closed-pore hTRPV6$_{GEN}$ results in one-helical turn shortening of the TRP helix and elongation of the unfolded region connecting S6 to the TRP helix (Fig. 3f). In subunits B and D of hTRPV6$_{GEN}$, the S6-TRP helix region also shows a drastic difference from the corresponding region in the 4-fold symmetrical structures. In this case, the TRP helix maintains approximately the same size as in the previously published closed-state structures, while α-helical S6 splits around M578 into two helical segments, S6 and S6-TRP (Fig. 3f, g and Supplementary Movie). Thus, both the S4–S5 and S6-TRP helix regions in hTRPV6$_{GEN}$ adapt the conformations different from all previously published structures of TRPV6.

### Genistein binding sites
Close inspection of the hTRPV6$_{GEN}$ cryo-EM map revealed two densities of the shape of a genistein molecule at the intracellular half of the

ion channel pore that were not present in the hTRPV6$_{Open}$ map (Fig. 4a, Supplementary Fig. 4). When fitted into these densities, both molecules of genistein appear to be located at the central pore axis and oriented with their long axis perpendicular to the pore axis. The upper site (1) is located above the gate, at the bottom of the central cavity, and is contributed by the S6 hydrophobic residues M570, L571 and L574 (Fig. 4b). The bottom site (2) is right below the gate, at the intracellular entrance to the ion channel pore, and contributed by the residues M578, G579, H582, W583 and A586 (Fig. 4c). The cryo-EM density at site 2 is somewhat weaker than at site 1. Since genistein was purified from a natural source, we cannot exclude the possibility that the lower site represents a bound contaminant. However, we believe that such a possibility is highly unlikely given the resemblance of the density with the molecule of genistein and the high degree (~99%) of the reagent purity, which was verified using multiple methods. CHS is also an unlikely candidate to represent the density in site 2 as this density does not resemble the shape of CHS and it only appears when genistein is added to the sample, while CHS is always present in all our TRPV6 preparations.

Due to the orientation of both genistein molecules perpendicular to the ion channel central axis, the two pairs of hTRPV6$_{GEN}$ diagonal subunits A/C and B/D contribute a different number of residues to genistein binding. Correspondingly, different strength of genistein interaction with the two pairs of diagonal subunits is the likely cause of

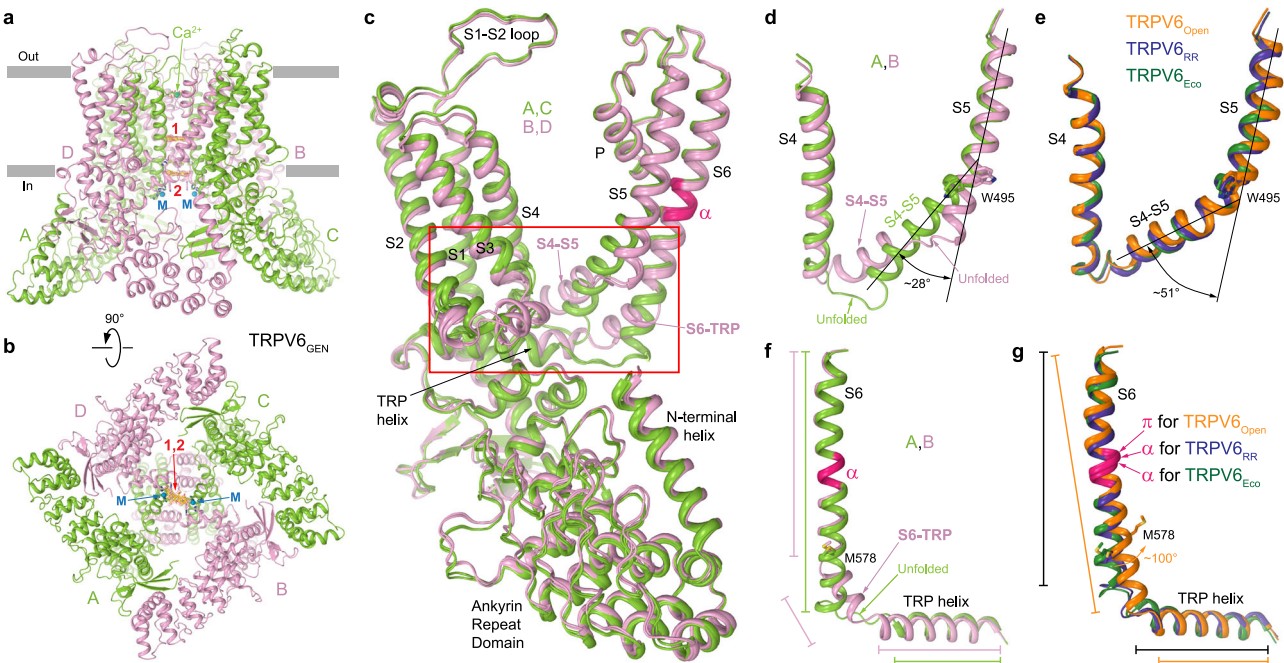

**Fig. 3 | Deviation of TRPV6_GEN from 4-fold rotational symmetry. a, b** TRPV6_GEN structure viewed parallel to the membrane (**a**) and intracellularly (**b**), with two diagonal subunits, A and C, colored green, another two, B and D, pink, and metal ion binding sites (M) shown as blue spheres. Molecules of genistein are shown as stick models and the corresponding cryo-EM density as red mesh. **c** Superposition of four TRPV6_GEN subunits. The red box outlines the region where subunits A and C show significant structural differences compared to subunits B and D. **d, e** Superposition of the S4–S5 regions in subunits A (green) and B (pink) of TRPV6_GEN (**d**) and TRPV6_Open (orange), TRPV6_RR (blue) and TRPV6_Eco (dark green) subunits (**e**). The angle between continuous helical regions S4–S5 and S5 is indicated. **f, g** Superposition of the S6-TRP helix regions in subunits A (green) and B (pink) of TRPV6_GEN (**f**) and TRPV6_Open (orange), TRPV6_RR (blue) and TRPV6_Eco (dark green) subunits (**g**). The region that undergoes the α-to-π transition in S6 is highlighted in hot pink. The lengths of helical regions and -100° rotation of S6 during channel opening are indicated.

asymmetric pore transformation (Fig. 3). This transformation appears to be stabilized by the formation of two asymmetrically located metal ion binding sites at the intracellular pore entry.

## Putative metal binding sites at the hTRPV6_GEN pore intracellular entry

Two strong densities at the pore intracellular entrance, located two-fold symmetrically relative to the central pore axis, are observed in the cryo-EM density of hTRPV6_GEN (Supplementary Fig. 6a). Each of these densities is in the middle of three-histidine clusters, which include H587 of the B/D diagonal subunits pair as well as H582 and H587 of the A/C diagonal subunits pair (Fig. 4a, c). The distances from the center of the density to histidines (Supplementary Fig. 6b, c) are 2.3–2.5 Å (H587 in B/D), 2.2–2.4 Å (H582 in A/C) and 2.2 Å (H587 in A/C), which are typical for histidine coordination of $Zn^{2+}$ ions[91–93], although other metal ions, such as $Mg^{2+}$, $Ca^{2+}$ or $Na^+$, can be coordinated as well[94–96]. These metal ions (M) are either the main components of our purification buffers ($Na^+$) or likely impurities of either the buffer components or genistein, which was extracted from *Styphnolobium japonicum*.

To test possible contribution of the putative metal ions to TRPV6 inhibition by genistein, we introduced alanine mutations of H582, H587, or both and used Fura-2 fluorescent measurements to estimate changes in calcium uptake through the corresponding mutant channels at different genistein concentrations (Fig. 4d). Genistein inhibition of calcium uptake through the H582A mutant was weaker than through wild-type channels as indicated by the rightward shift of the genistein concentration-dependence. A stronger shift in the concentration-dependence was observed for H587A and H582A/H587A mutants, with a little difference between the two, consistent with the predominant contribution (two out of three) of H587 to the putative metal ion coordination (Fig. 4a, c and Supplementary Fig. 6). Therefore, our mutagenesis combined with functional recordings are consistent with

the putative metal ion coordination by the three-histidine clusters being an additional factor that might help to stabilize the asymmetric conformation of hTRPV6_GEN.

## MD simulations of genistein binding sites

To further validate genistein binding to sites 1 and 2, we carried out all-atom MD simulations (Fig. 5, Supplementary Figs. 7-8). Four simulations were performed for site 1 with different initial orientations of the ligand, which differed by 180° rotations around the long genistein axis and around the central pore axis. Our simulations showed that the ligand is stabilized in site 1 by two hydrogen bonds between the hydroxyl groups of genistein and the carbonyl oxygens of M570 in subunits A/C (Supplementary Fig. 7b). These interactions orient the ligand in either one of two symmetrical positions rotated by 180° around the central pore axis, thus demonstrating an excellent fit to the non-protein cryo-EM density in site 1 (Fig. 5a). Similar simulations were repeated with CHS embedded in site 1. CHS showed no specific interactions with the protein, was much more mobile than genistein, and intended to escape from site 1. The MD-predicted density for CHS formed a "band" across the pore, inconsistent with the cryo-EM density (Supplementary Fig. 8a, c).

MD simulations of a single genistein molecule placed at site 2 starting with the position modeled in the cryo-EM structure (Fig. 4c) revealed highly dynamic behavior of this molecule, only somewhat restrained by π-stacking interactions with W583 (not shown). To stabilize the genistein molecule at this cryo-EM-like primary position, we added another genistein molecule at the secondary position between the W583 side chains below and perpendicular to the primary position (Supplementary Fig. 7a). We performed four MD simulations with different initial orientations of genistein molecules in the primary and secondary positions at site 2, generated in the same way as in simulations for site 1. The simulations showed that genistein in the primary

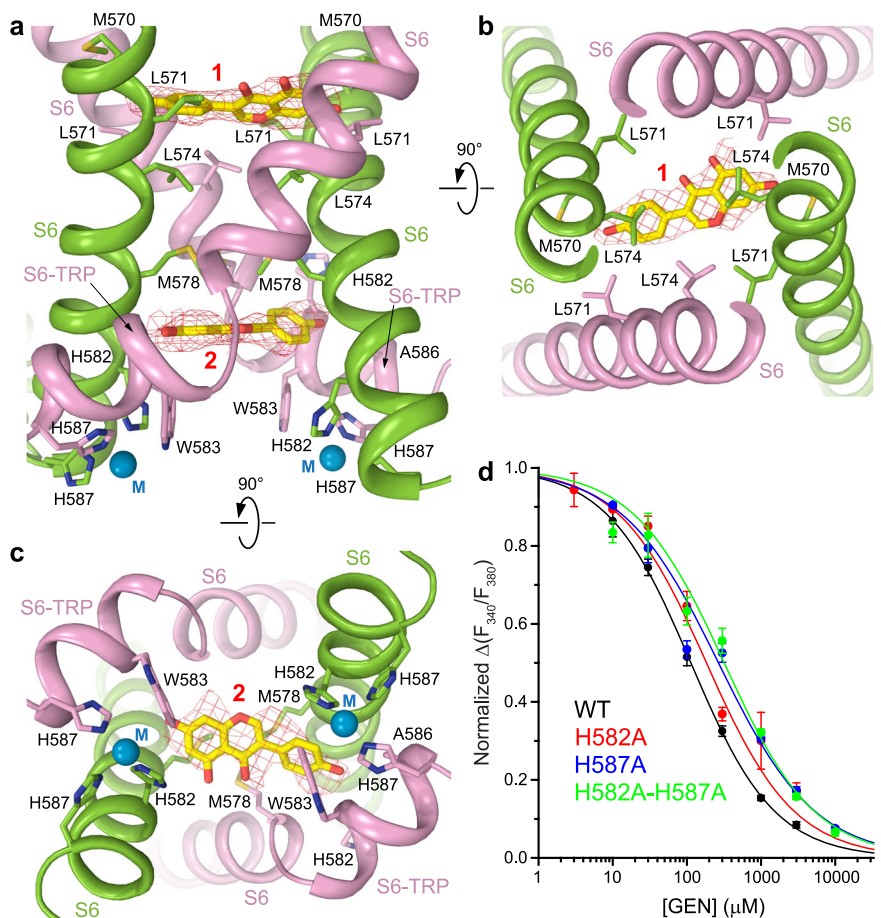

**Fig. 4 | Binding sites of genistein and metal ions. a** Side view of genistein binding sites 1 and 2 in the TRPV6$_{GEN}$ structure, with subunits A and C colored green, B and D pink, and metal ions (M) shown as blue spheres. Molecules of genistein and side chains that contribute to genistein or metal ion binding are shown in sticks and the cryo-EM density for genistein (in stick models) as red mesh. **b**, **c** Intracellular views of genistein binding sites 1 (**b**) and 2 (**c**). **d** Dose-response curves for genistein inhibition of Ca$^{2+}$ uptake through wild-type (black) and H582A (red),

H587A (blue) and H582A-H587A (green) mutant TRPV6 channels. Curves through data points are fitted with the logistic equation, with the mean ± SEM values of $IC_{50}$ and $n_{Hill}$, 113 ± 5 µM and 0.77 ± 0.02 for wild-type ($n = 3$ independent experiments), 184 ± 24 µM and 0.73 ± 0.04 for H582A ($n = 3$ independent experiments), 267 ± 29 µM and 0.67 ± 0.03 for H587A ($n = 3$ independent experiments), and 318 ± 45 µM and 0.71 ± 0.06 for H582A-H587A ($n = 3$ independent experiments).

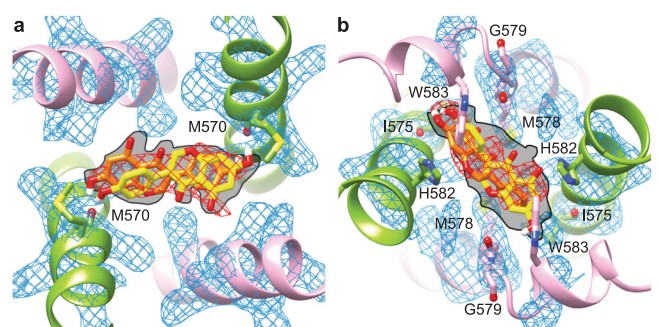

**Fig. 5 | Molecular dynamics simulation of genistein binding to sites 1 and 2. a**, **b** Extracellular view of genistein at site 1 (**a**) and intracellular view of genistein at the primary position in site 2 (**b**). Gray filling shows the density for genistein heavy atoms averaged over the MD runs. The experimental cryo-EM protein and non-protein densities are shown as blue and red mesh, respectively. Protein subunits A/C (green) and B/D (pink) are shown as cartoon models, with residues forming hydrogen bonds and π-stacking interactions with genistein shown in sticks and labeled. The most populated MD states of genistein are illustrated by representative yellow and orange stick models.

position of site 2 is more mobile than genistein in site 1. However, in three out of four MD runs, genistein was stabilized in the primary position of site 2 by π-stacking interactions with W583 in subunits B/D or H582 in subunits A/C as well as by one or two hydrogen bonds with the carbonyl oxygens of the neighboring residues M578 and G579 in subunits A/C or residue I575 in subunits B/D (Supplementary Fig. 7c). Such a diversity of interaction partners did not allow to reveal specific positions of the ligand in the site. However, the resulting distribution of the ligand density at the primary position in site 2 averaged over the MD runs demonstrated excellent agreement with the central non-protein cryo-EM density (Fig. 5b).

Genistein at the secondary position in site 2 was much more mobile than the genistein molecule at site 1 or in the primary position at site 2 (Supplementary Fig. 7d). Nevertheless, the genistein molecule at the secondary position in site 2 tended to form two to four π-stacking interactions with W583 in subunits B/D and H582 in subunits A/C as well as hydrogen bonds with the side chains of D580 in subunits B/D. In turn, these interactions imposed restraints on the genistein molecule at the primary position in site 2. The cumulative behavior of two genistein molecules at site 2 agrees well with the cryo-EM data, where a group of small non-protein densities between W583 side chains are likely to represent the ensemble of positions of the weakly

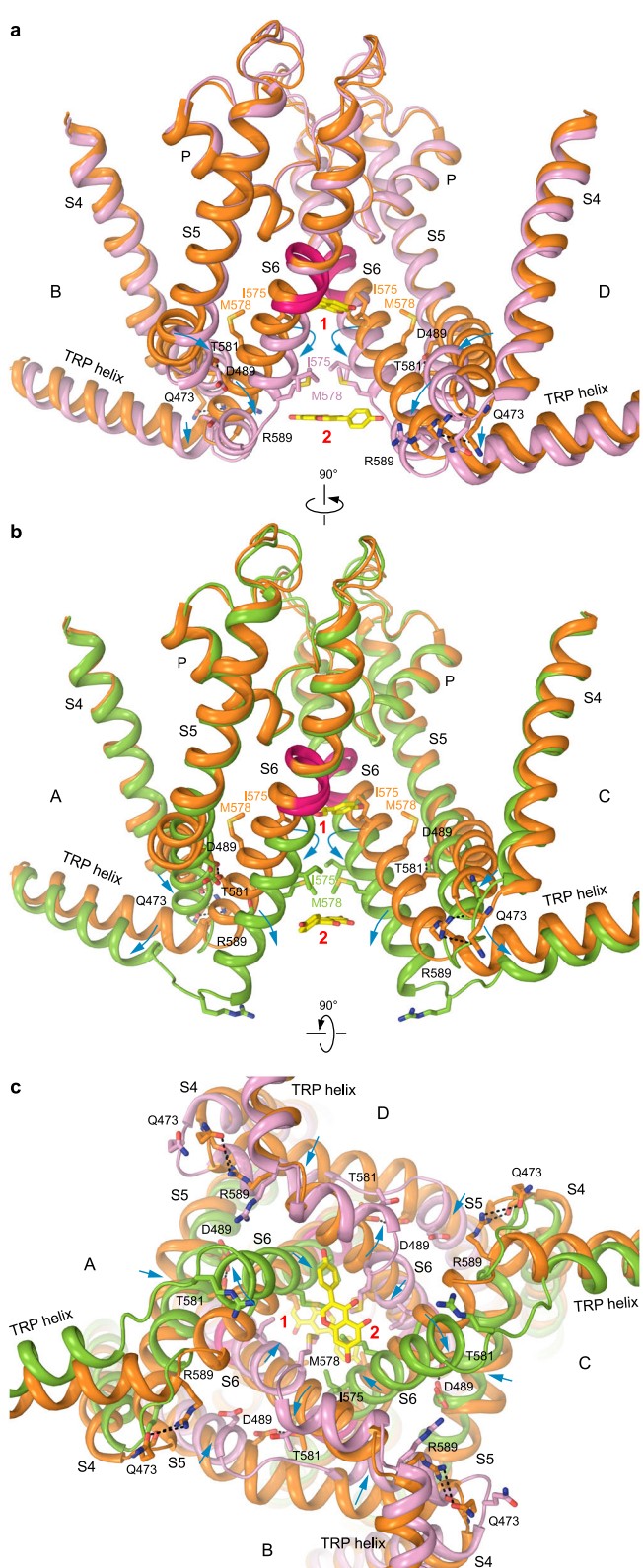

**Fig. 6 | Molecular mechanism of TRPV6 inhibition by genistein.**
**a**–**c** Superposition of the pore region in TRPV6$_{Open}$ (orange) and TRPV6$_{GEN}$ viewed parallel to the membrane (**a**, **b**) or intracellularly (**c**), with subunits A and C colored green, B and D pink. Only two of the four subunits are shown in **a** and **b**, with the front and back subunits omitted for clarity. Molecules of genistein (in stick model) and TRPV6 residues that contribute to the pore narrow constriction or open state-stabilizing interactions are shown in sticks. Dashed lines indicate interactions between D489 and T581 as well as Q473 and R589 in the open state. Blue arrows indicate the movement of domains that accompany closure of the pore upon genistein binding.

density revealed by cryo-EM (Supplementary Fig. 8b). Accordingly, while there is a possibility that the experimentally observed non-protein density in site 2 represents CHS or a similar size small molecule, the probability of this is low.

## Discussion

The superposition of hTRPV6$_{Open}$ and hTRPV6$_{GEN}$ structures suggests a possible mechanism of TRPV6 inhibition by genistein (Fig. 6 and Supplementary Movie). When binding to the open TRPV6 channel perpendicularly to the central pore axis, genistein molecules pull one diagonal pair of subunits (A/C) asymmetrically towards the channel center. This motion breaks the interactions between D489 in S5 and T581 in S6 as well as Q473 in the S4–S5 linker and R589 in the TRP helix, which stabilize the energetically unfavorable α-to-π transition in S6 and the open-pore conformation, respectively[87]. The reverse π-to-α transition in S6 is accompanied by a -100° rotation of the intracellular portion of S6, while the stabilizing interactions with genistein molecules allow S6 in subunits A and C to stay as long as in the open state and not become shorter as in the previously published closed-state structures[27,87,90]. This in turn causes shortening of the TRP helices in subunits A and C due to unwinding of their N-terminal portions (Fig. 6b, c).

Upon genistein binding, the pore-forming domains in subunits B and D also change their conformations to adapt to changes in subunits A and C. This adaptation disrupts the D489-T581 and Q473-R589 interactions, reverses the unfavorable α-to-π transition in S6 but also causes the split of S6 into two helical segments, S6 and S6-TRP (Fig. 6a, c). In turn, the S4–S5 regions which stay in direct contact with the conformationally altered regions of S6-TRP helix also adapt their conformation. In subunits A and C, this involves altering the angle between S4–S5 and S5 and increasing the length of the unfolded connection between S4 and S4–S5. In contrast, the contiguous helical region that follows S4 in subunits B and D splits into two distinct helical segments, S4–S5 and S5 (Fig. 3d and 6). Amazingly, these genistein-induced conformational changes are localized to the channel intracellular core, do not propagate beyond the S4–S5 and S6-TRP helix regions, and the rest of the TRPV6 molecule remains essentially the same (Fig. 3a, b). Currently, it is difficult to assess the relative contribution of sites 1 and 2 to the mechanism of TRPV6 inhibition by genistein. Given the more stable behavior of genistein at site 1 compared to site 2, where a more mobile genistein molecule at the primary central position might be accompanied by an even more mobile genistein molecule at the secondary position (Fig. 5, Supplementary Fig. 7), as well as a relatively weak contribution of metal coordination in the vicinity of site 2 to inhibition of calcium influx (Fig. 4d), we hypothesize that site 1 represents the main site of genistein action.

So far, 14 unique ligand binding sites have been identified in TRPV channels[28]. Genistein binds to the intracellular pore entry site, which makes it a TRPV6 ion channel blocker. Given the small size of the TRPV6 selectivity filter compared to the genistein molecule, genistein likely approaches its sites in the intracellular pore by first crossing through the membrane. In fact, the relatively low affinity of TRPV6 to genistein (Fig. 1c, 4d) may be due to a slow crossing of the drug through the membrane to reach the intracellular side. Given the

interacting ligands (Supplementary Fig. 7a). MD simulations with CHS embedded in the primary position of site 2 in four different initial orientations revealed that the ligand is stabilized in this site much better than in site 1 (Supplementary Fig. 8b, d). Pockets between the S6 helices of neighboring subunits provide enough space for the CHS molecule, while its acidic group forms polar interactions with K484 or R589. However, being tightly packed in site 2, the CHS molecule created a much more extended density in MD simulations than the

location of site 1 above the channel gate formed by the S6 bundle crossing and at the bottom of the central cavity, genistein is likely to reach this site only after the channel undergoes opening. If this is true, genistein can be considered an open-channel blocker. It does not, however, interact passively with the ion channel as the majority of open-channel blockers do. Instead, because of the non-equivalent positioning of genistein molecules in the pore relative to two pairs of diagonal subunits (Fig. 4), they cause asymmetric transformation of the pore (Fig. 3, Supplementary Movie) and its closure (Fig. 2).

What makes genistein a unique ion channel blocker of TRPV6? Among 14 types of TRPV ligands that were characterized structurally[28], there are four types of ion channel blockers. TRPV6 can be blocked by trivalent cations, like $Gd^{3+}$, which bind at the extracellular pore entry site formed by side chains of four D542 residues, each from four individual TRPV6 subunits. At this location, $Gd^{3+}$ binding occludes the pore for conductance but does not change the conformation of the selectivity filter, which is stabilized by hydrophobic interactions of the pore helix residues, including phenylalanines F531, F534, and F537, neither does it alter the conformation of the rest of TRPV6[86,89]. RR represents the second type of ion channel blockers, which bind to the selectivity filter site located intracellularly with respect to the extracellular pore entry site[27]. Binding of RR at this location of the TRPV6 pore does not introduce significant conformational changes in the selectivity filter either but causes a conversion of the gate region into the closed-state, presumably due to electrostatic interactions of the positive charge of RR and the electric dipole of the S6 helix[27].

The other two types of ion channel blockers bind at the pore intracellular entry site, the region where genistein binds as well. The physiological blocker calmodulin (CaM) interacts with this site through a unique cation-π interaction by inserting the side chain of lysine K115 into a tetra-tryptophan (W583) cage at the pore's intracellular entrance[26,97]. Similarly, (4-phenylcyclohexyl)piperazine derivatives (PCHPDs), selective nanomolar-affinity synthetic inhibitors of TRPV6 including cis-22a, plug the channel pore at the pore intracellular entry site, mimicking the action of CaM[26]. These two types of TRPV6 ion channel blockers induce conformational rearrangements that transform the open pore into a non-conducting inactivated state, different from the closed and open conformations of TRPV6. In the inactivated state, the TRPV6 pore becomes hydrophobically sealed by the I575 side chains, while the π-bulge remains present in the middle of S6, similar to the open state[26,97]. At the same time, one of the open state-stabilizing interactions breaks (Q473-R589), but the other one remains intact (D489-T581).

Similar to PCHPDs and CaM, genistein binds at the intracellular pore entry but causes drastically different structural rearrangements, accompanied by alterations of the local symmetry and secondary structure of the S4–S5 and S6-TRP regions (Fig. 3 and Supplementary Movie). The two likely reasons for such dramatic rearrangements are (1) flexibility of the pore intracellular region, which has to easily permit gate opening and closure to maintain TRPV6 constitutive activity, and (2) the orientation of genistein molecules across the pore instead of along the pore, as in the case of PCHPDs and CaM. The latter factor allows genistein to cause more damaging mechanistic changes in the structure of TRPV6 pore compared to any other ion channel blocker studied before. Due to the unprecedented character of genistein-induced conformational transformation of the channel, the mechanism of genistein block is also unique amongst all previously described mechanisms of TRP channel inhibition by various natural and synthetic antagonists[28], including allosteric inhibition of hTRPV6 by 2-APB or econazole. With the potential to utilize various well-established chemical synthesis pathways for genistein and its derivatives[31,98,99], the mechanism of ion channel block described here opens new avenues for the development of drugs targeting TRPV6 in pathological conditions.

## Methods

### Construct
Full-length wild-type human TRPV6 used for cryo-EM was cloned into a pEG BacMam vector[100] with a C-terminal thrombin cleavage site followed by a streptavidin affinity tag (WSHPQFEK). For Fura-2 AM measurements, point mutations in wild-type human TRPV6 were introduced using the standard molecular biology techniques[27,101].

### Expression and purification
hTRPV6 was expressed and purified based on our previously established protocols[27,102–104]. Bacmids and baculoviruses were produced using standard procedures[26,27,100,102]. Baculovirus was made in Sf9 cells for ~72 h (Thermo Fisher Scientific, mycoplasma test negative, GIBCO #12659017) and was added to suspension-adapted HEK 293S cells lacking N-acetyl-glucosaminyltransferase I (GnTI⁻, mycoplasma test negative, ATCC #CRL-3022) that were maintained at 37 °C and 5% $CO_2$ in Freestyle 293 media (Gibco-Life Technologies #12338-018) supplemented with 2% FBS. Twenty-four hours after transduction, 10 mM sodium butyrate was added to enhance protein expression, and the temperature was reduced to 30 °C. Seventy-two hours after transduction, cells were harvested by centrifugation at $5471 \times g$ for 15 min using a Sorvall Evolution RC centrifuge (Thermo Fisher Scientific), washed in phosphate-buffered saline pH 8.0, and pelleted by centrifugation at $3202 \times g$ for 10 min using an Eppendorf 5810 centrifuge. The cell pellet was solubilized under constant stirring for 2 h at 4 °C in ice-cold lysis buffer containing 1% (w/v) n-dodecyl β-D-maltoside, 0.1% (w/v) CHS, 20 mM Tris-HCl pH 8.0, 150 mM NaCl, 0.8 μM aprotinin, 4.3 μM leupeptin, 2 μM pepstatin A, 1 mM phenylmethylsulfonyl fluoride, and 1 mM β-mercaptoethanol (βME). The non-solubilized material was pelleted in the Eppendorf 5810 centrifuge at $3202 \times g$ and 4 °C for 10 min. The supernatant was subjected to ultracentrifugation in a Beckman Coulter ultracentrifuge using a Beckman Coulter Type 45Ti rotor at $186,000 \times g$ and 4 °C for 1 h to further clean up the solubilized protein. The supernatant was added to 5 ml of strep resin and rotated for 1 h at 4 °C. The resin was washed with 10 column volumes of wash buffer containing 20 mM Tris-HCl pH 8.0, 150 mM NaCl, 1 mM βME, 0.01% (w/v) GDN, and 0.001% (w/v) CHS, and the protein was eluted with the same buffer supplemented with 2.5 mM d-Desthiobiotin. The eluted protein was concentrated using a 100 kDa NMWL centrifugal filter (MilliporeSigma Amicon) to 0.5 ml and then centrifuged in a Sorvall MTX 150 Micro-Ultracentrifuge (Thermo Fisher Scientific) using an S100AT4 rotor for 30 min at $66,000 \times g$ and 4 °C before being injected into a size-exclusion chromatography (SEC) column. hTRPV6 was further purified using a Superose™ 6 10/300 GL SEC column attached to an AKTA FPLC (GE Healthcare) and equilibrated in 150 mM NaCl, 20 mM Tris-HCl pH 8.0, 1 mM βME, 0.01% GDN, and 0.001% CHS. The tetrameric peak fractions were pooled and concentrated using 100 kDa NMWL centrifugal filter to 3.36 mg/ml. Genistein (2 mM) was added to hTRPV6 and the resulting sample was incubated at room temperature for 120 min before grid freezing.

### Cryo-EM sample preparation and data collection
UltrAuFoil R 1.2/1.3, Au 300 grids were used for plunge-freezing. Prior to sample application, grids were plasma treated in a PELCO easiGlow glow discharge cleaning system (0.39 mBar, 15 mA, "glow" 25 s, "hold" 10 s). A Mark IV Vitrobot (Thermo Fisher Scientific) set to 100% humidity at 4 °C was used to plunge-freeze the grids in liquid ethane after applying 3 μl of protein sample to their gold-coated side using a blot time of 5 s, a blot force of 5, and a wait time of 15 s. The grids were stored in liquid nitrogen before imaging.

Images of frozen-hydrated particles of hTRPV6 in the presence of genistein were collected using Leginon[105–107] software on a Titan Krios transmission electron microscope (Thermo Fisher Scientific) operating at 300 kV and equipped with a post-column GIF Quantum energy

filter and a Gatan K3 Summit direct electron detection camera (Gatan, Pleasanton, CA, USA). 3630 micrographs were collected in counting mode with a raw image pixel size of 0.83 Å across the defocus range of −0.8 to −2.0 μm. The total dose of ~60 e⁻Å⁻² was attained by using the dose rate of ~16 e⁻ pixel⁻¹ s⁻¹ across 50 frames during the 2.5-s exposure time.

## Image processing and 3D reconstruction

Data were processed in RELION[108] and cryoSPARC[109]. Movie frames were aligned using the RELION's implementation of a MotionCor2[110]-like algorithm. Contrast transfer function (CTF) estimation was performed on non-dose-weighted micrographs using the patch CTF estimation in cryoSPARC. Subsequent data processing was done on dose-weighted micrographs. Following CTF estimation, micrographs were manually inspected and those with outliers in defocus values, ice thickness, and astigmatism as well as micrographs with lower predicted CTF-correlated resolution (higher than 5 Å) were excluded from further processing (individually assessed for each parameter relative to the overall distribution). After several rounds of selection through 2D classification, particles were further 3D classified (heterogeneous refinement) into four classes. Particles representing the best class were re-extracted without binning (256-pixel box size) and further 3D classified.

The final sets of particles for hTRPV6$_{GEN}$ and hTRPV6$_{Open}$ representing the best classes were subjected to homogeneous refinement. The reported resolutions of 2.66 Å and 2.71 Å for hTRPV6$_{GEN}$ and hTRPV6$_{Open}$, respectively, were estimated using the gold standard Fourier shell correlation (GSFSC) (Supplementary Figs. 2–3). The local resolution was calculated with the resolution range estimated using the FSC = 0.143 criterion. Cryo-EM density visualization was done in UCSF Chimera[111] and UCSF ChimeraX[112].

## Model building

Models of hTRPV6$_{GEN}$ and hTRPV6$_{Open}$ were built in Coot[113], using the previously published cryo-EM structure of TRPV6 in the open state (PDB ID: 7S89)[27] as a guide. The models were tested for overfitting by shifting their coordinates by 0.5 Å (using Shake) in Phenix[114], refining the shaken models against the corresponding unfiltered half maps, and generating densities from the resulting models in UCSF Chimera. Structures were visualized and figures were prepared in UCSF Chimera, UCSF ChimeraX, and Pymol[115]. The pore radius was calculated using HOLE[116].

## Fura-2 AM measurements

Full-length wild-type or mutant human TRPV6 was expressed in HEK 293S cells as described above. 48 h after transduction, the cells were harvested by centrifugation at 550 × g for 5 min, resuspended in pre-warmed HEPES-buffered saline (HBS: 118 mM NaCl, 4.8 mM KCl, 1 mM MgCl₂, 5 mM D-glucose, 10 mM HEPES pH 7.4) containing 5 μg/ml of Fura-2 AM (Life Technologies) and incubated at 37 °C for 45 min. The loaded cells were then centrifuged for 5 min at 550 × g, resuspended again in prewarmed HBS, and incubated at 37 °C for 30 min in the dark. The cells were subsequently pelleted and washed twice, then resuspended in HBS. The cells were kept on ice in the dark for a maximum of ~2 h before fluorescence measurements, which were conducted using a spectrofluorometer QuantaMaster 40 (Photon Technology International) at room temperature in a quartz cuvette under constant stirring. Intracellular Ca²⁺ was measured by taking the ratio of fluorescence measurements at two excitation wavelengths (340 and 380 nm) and one emission wavelength (510 nm). The excitation wavelength was switched at 200-ms intervals.

## Electrophysiology

DNA encoding wild-type human TRPV6 was introduced into a plasmid for expression in eukaryotic cells that was engineered to produce GFP

via a downstream internal ribosome entry site[117]. HEK 293S cells (ATCC #CRL-1573) grown on glass coverslips in 35-mm dishes were transiently transfected with 1–5 μg of plasmid DNA using Lipofectamine 2000 Reagent (Life Technologies). Recordings were made 24 h after transfection at room temperature. Currents from whole cells, typically held at a 0-mV potential, were recorded using an Axopatch 200B amplifier (MolecularDevices, LLC), filtered at 5 kHz, and digitized at 10 kHz using low-noise data acquisition system Digidata 1440 A and pCLAMP software (Molecular Devices, LLC). The external solution contained (in mM): 142 LiCl, 10 HEPES, and 10 glucose, pH 7.4. To evoke monovalent currents, 0.1–0.5 mM EGTA was added to the external solution. The internal solution contained (in mM): 100 CsAsp, 20 CsF, 10 EGTA, 3 MgCl₂, 4 NaATP, and 20 HEPES pH 7.2, an additional 1 mM ATP was added immediately before the experiment. We used the LiCl-based extracellular solution, because, in this solution, removal of extracellular Mg²⁺ and Ca²⁺ does not induce endogenous currents in non-transfected cells, unlike in Na⁺ or K⁺ based solutions[118]. TRPV6 currents were recorded in response to 400-ms voltage ramps from −100 mV to +70 mV applied every 5–10 s. Genistein was added directly to the aqueous buffer solutions for measurements in HEK cells. At concentrations higher than 50 μM, genistein displayed aggregation behavior and tended to clog our application system. For this reason, all experiments were carried out at genistein concentrations lower than 50 μM. Data analysis was performed using the computer program Origin 9.1.0 (OriginLab Corp.).

## MD simulations

The model of TRPV6$_{GEN}$ (residues 27–638) was inserted into a hydrated lipid bilayer with the molecular composition of 50% palmitoyloleoyl-phosphatidylcholine (POPC), 25% palmitoyloleoylphosphatidylethanolamine (POPE), and 25% cholesterol (about 900 lipids in total). Ca²⁺ ion was embedded into the selectivity filter region (between the side chains of residues D542) and two Zn²⁺ ions into the metal binding sites like in the structural model. Na⁺ and Cl⁻ ions were added to ensure zero net charge of the system at 0.15 M ionic concentration. Twelve replicas of the system were constructed: four replicas with a differently oriented genistein in site 1, four replicas with two differently orientated genistein molecules at the primary and secondary positions in site 2, and four replicas with two differently orientated CHS molecules in sites 1 and 2 (see the text and Supplementary Table 2 for details).

To stabilize the protein structure in the vicinity of the binding sites, intersubunit constraints were applied to the distances between C$_\alpha$ atoms of the lower parts of the S6 helices. The distances were constrained between each residue in the region 563–578 (536–586 for the systems with genistein molecules in the primary and secondary positions in site 2) of all four subunits with the force constant of 10 kJ/(mol × Å²). During MD simulations, such a network of constraints provides the necessary flexibility for protein adaptation to the ligand but retains the binding site structure close to the cryo-EM model.

First, the simulated systems were equilibrated in several stages: 5 × 10³ steps of steepest descent minimization followed by heating from 5 to 310 K during a 100 ps MD run, then 10 ns of MD run with fixed positions of the protein and genistein/CHS heavy atoms, 10 ns of MD with fixed positions of the protein backbone and genistein/CHS heavy atoms, 20 ns of MD with fixed positions of the protein Cα atoms to permit membrane and genistein/CHS relaxation after insertion. Finally, 200 ns MD-production runs were carried out.

MD simulations were carried out using GROMACS 2021.4 package[119], CHARMM36 force field[120–125], and the TIP3P water model[126]. Simulations were performed with an integration time of 2 fs, constrained hydrogen-containing bond lengths by the LINCS algorithm[127], imposed 3D periodic boundary conditions, at constant temperature (310 K) and pressure (1 bar). A cutoff distance of 12 Å was used to evaluate nonbonded interactions and the particle-mesh Ewald

method[128] was employed to treat long-range electrostatics. The CHARMM36 topology for genistein was taken from Burendahl et al.[129].

## Reporting summary

Further information on research design is available in the Nature Portfolio Reporting Summary linked to this article.

## Data availability

All data needed to evaluate the conclusions of the paper are present in the paper or the Supplementary Information. The cryo-EM density map of hTRPV6 in the open state and in complex with genistein were deposited to the Electron Microscopy Data Bank (EMDB) under the accession codes EMD-29343 (hTRPV6$_{GEN}$) and EMD-29344 (hTRPV6$_{Open}$). The atomic coordinates have been deposited to the Protein Data Bank (PDB) under the accession codes 8FOA [https://doi.org/10.2210/pdb8FOA/pdb] (hTRPV6$_{GEN}$) and 8FOB [https://doi.org/10.2210/pdb8FOB/pdb] (hTRPV6$_{Open}$) (see also Supplementary Table 1). The coordinates of the protein and ligands obtained in MD simulations are available as supplementary pdb files with the step of 20 ns (including the initial configuration at $t = 0$, see also Supplementary Table 2). All other data are available from the corresponding author upon request. Source data are provided with this paper.

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

## Acknowledgements

We thank Robert Grassucci, Zhening Zhang, Surajit Banerjee, and Yen-Hong Kao (Columbia University Cryo-Electron Microscopy Center) for helping with microscope operation and data collection. This work was performed at the Columbia University Cryo-Electron Microscopy Center. Access to computational facilities of the Supercomputer Center "Polytechnical" at the St. Petersburg Polytechnic University is greatly appreciated. A.N. is a Walter Benjamin Fellow funded by the Deutsche Forschungsgemeinschaft (DFG, German Research Foundation) – 464295817. M.D. simulations were supported by the RSF (23-14-00313). A.I.S. was supported by the NIH (R01 CA206573, R01 AR078814, R01 NS083660, and R01 NS107253).

## Author contributions

A.N. made constructs and carried out protein expression, protein purification, and cryo-EM data processing. M.V.Y. carried out electrophysiology experiments. A.N. and K.D.N. prepared cryo-EM samples and carried out Fura-2 intracellular calcium imaging recordings. A.N. and A.I.S. analyzed structural data and built molecular models. Y.A.T., N.A.K., and R.G.E. designed computational work, performed molecular modeling, and analyzed the data. A.N. and A.I.S. wrote the manuscript, which was then edited by all authors.

## Competing interests

The authors declare no competing interests.
