## [Peer Review File · Nature Communications]

REVIEWER COMMENTS

Reviewer #1 (Remarks to the Author):

TRPV6 is an ion channel involved in many physiological processes and pathological conditions through its contribution to intracellular calcium homeostasis. The Sobolevsky group has previously reported several important TRPV6 structures that have helped to lay the foundation for mechanistic understanding of TRPV6 function. The present manuscript describes a new structure of TRPV6 in complex with genistein, which is known for inhibiting the channel. They showed the binding stoichiometry to be two per channel, the binding location to be at the intracellular pore entrance (distinct from other TRPV6 binding molecules/ions, as revealed by structural studies done previously in the same lab), and genistein binding is associated with noticeable conformational rearrangements that appear to mirror the open to closed gating transition. It is noticed that conformational rearrangements are localized to the pore region. These findings reveal a detailed structural basis for genistein-TRPV6 interaction, which has been actively investigated for potential clinical applications. Several suggestions for further improvement of this exciting study are discussed below.

I wonder if the way genistein works still qualifies it as a channel “blocker” as it is referred to in the manuscript. The two bound genistein molecules are likely blocking ion permeation at the observed positions, however as the authors described, the channel pore undergoes a conformational change, from the open conformation to one that is very similar to the known closed channel conformation (S6 being alpha-helix, very small pore diameter, etc.). Therefore, genistein seems to act as a gating modifier that happens to bind at the pore entrance.

Two genistein molecules are identified per TRPV6 channel in a unique combination of poses. Given the conformational change these two genistein molecules induce, it is expected that their binding would show cooperativity. However, the function data showed inhibition in an extremely wide, four orders of magnitude concentration range. This observation is puzzling and not fully consistent with the structural model. It is possible calcium imaging, being an indirect measurement, does not faithfully report inhibition of channel activity. If the authors believe this is the case, then they should confirm with patch clamp which they have used in the present and previous studies, and to fit inhibition of channel currents to a Hill (instead of logistic) equation to obtain the Hill slope estimate.

Significance of metal binding is suggested in the manuscript, but not well established. If the bound metal ions came from the protein preparation stages, as the authors speculated, what might be the physiological metal ions? Would their binding facilitate conformational change to the closed-like one that genistein binding produced (point mutations yielded rather small shifts of the very shallow inhibition curve)? If it is considered physiological, then won't the previously reported closed channels (hTRPV6 in different lipid compositions, rat TRPV6 channels, etc.) contain the bound metal ions as well?

Finding the identity of the bound metal ions, for example using inductively coupled plasma mass spectrometry (ICP-MS) or X-ray energy scan, would be very helpful. Directly testing effects of adding metal ions (would require patch clamp in inside-out configuration) would be even better. In the absence of additional information, it is suggested to de-emphasize this part and make the associated discussion conservative.

The term “straight line” should be changed to “curve” (figure legend).

Jie Zheng

Reviewer #3 (Remarks to the Author):

The paper by Neuberger et al. reported the potential TRPV6 binding sites of genistein, a natural blocker, and the structural changes induced by the binding events. Two binding sites were assigned to the ligand above and below the crossing points of four S6 helices. The cryo-EM structures depicted symmetry breakdown and conformational changes in the specific regions close to the intracellular entry site (S2), which are pretty interesting and make genistein a different type of blockers that might have potential for structure-based drug design should its binding affinity be improved to nM range and be more specific. The paper is of unique interest to the field in both structural studies of ligands of TRP channels and the cryo-EM structure-based drug design. A few points need to be addressed before the paper is ready for publication.

1) The two binding sites were assigned without showing details of the ligand densities, especially the matching of the ligand models with the density shape and the bonding structures for stabilizing the ligands in the two binding pockets. As a potential compound of clinical applications, it is especially important to specify if the ligand densities are resolved sufficiently well to define unique orientations and configurations (around totable bonds) of the ligands in the two sites.

2) From a medicinal standpoint, the genistein affinity is relatively low using the two assays reported in the paper and it appears to lack specificity given the various applications. But the binding in S1 / S2 sites in the structure and the described conformational changes appear to suggest strong cooperativity or synergy, especially in consideration of no structure showing only one site (likely S1) occupied by genistein. Is the low affinity due to the application of genistein from the extracellular side and the slow crossing of genistein through the membrane? It is better to measure genistein affinity in a competition assay (on-off at the S2), and the channel block and time course of its action by applying it to channels in

inside-out patches (more directly on the S1 due to blocking the channel). These assays may help differentiate the two sites.

3) As a natural product, what contaminants are there in genistein? In what molar fractions? What solvent was used for purification? If a contaminant of less than 0.5% in abundance, but has a pretty high affinity, say $K_d \sim 100$ nM, it might compete for at least one of the assigned binding sites and change the story significantly.

4). Why was it necessary to incubate genistein with the proteins for 2 hours before freezing? In cell incubation it takes time probably for genistein to cross membranes, but not in purified proteins. Was the result the same if the incubation was brief?

5) Because of the promiscuity from two binding sites, are the ligand densities sufficiently well resolved to differentiate the rings of the flavone backbone in both sites from the ring structure of CHS, given the abundance of the latter?

6) Given the membrane permeability of genistein, is it possible that it is enriched in detergent micelles and exerts its effect from the periphery next to the S45 linker and the S6-TRP helix?

7) The term “intracellular entrance” is more suitable for the location of the S2 site, but not so for the S1. It is better to differentiate them in description because of their physical difference.

8) The mechanistic description of the morphed conformational changes in a stepwise fashion in the text and in Fig. 5 is highly speculative and might be misleading without other data. It is better to shorten it or put it in discussion.

9) Given the equilibrium in ligand binding to the proteins in solution, can the populations of the different structures be used to deduce energetic scale for genistein binding (likely at S2)? Or can the interaction energy between genistein and the two binding pockets be calculated and differentiate them?

We are very thankful to Reviewers for their excellent suggestions. We have made changes in the manuscript accordingly with the details outlined in our responses below.

REVIEWER COMMENTS

Reviewer #1 (Remarks to the Author):

TRPV6 is an ion channel involved in many physiological processes and pathological conditions through its contribution to intracellular calcium homeostasis. The Sobolevsky group has previously reported several important TRPV6 structures that have helped to lay the foundation for mechanistic understanding of TRPV6 function. The present manuscript describes a new structure of TRPV6 in complex with genistein, which is known for inhibiting the channel. They showed the binding stoichiometry to be two per channel, the binding location to be at the intracellular pore entrance (distinct from other TRPV6 binding molecules/ions, as revealed by structural studies done previously in the same lab), and genistein binding is associated with noticeable conformational rearrangements that appear to mirror the open to closed gating transition. It is noticed that conformational rearrangements are localized to the pore region. These findings reveal a detailed structural basis for genistein-TRPV6 interaction, which has been actively investigated for potential clinical applications. Several suggestions for further improvement of this exciting study are discussed below.

I wonder if the way genistein works still qualifies it as a channel “blocker” as it is referred to in the manuscript. The two bound genistein molecules are likely blocking ion permeation at the observed positions, however as the authors described, the channel pore undergoes a conformational change, from the open conformation to one that is very similar to the known closed channel conformation (S6 being alpha-helix, very small pore diameter, etc.). Therefore, genistein seems to act as a gating modifier that happens to bind at the pore entrance.

The Reviewer is absolutely right in that genistein, being an ion channel blocker (because it occludes the permeation pathway), also works as a gating modifier of hTRPV6. Throughout the history of ion channel block studies, it has been recognized relatively early that ion channel blockers can be crudely divided into those that interfere with gating (for example, sequential or foot-in-the door blockers) and those that do not (trapping blockers). For instance, in our previous study on the inhibition of hTRPV6 by the blocker ruthenium red (Neuberger *et al.* 2021), we found that binding of ruthenium red (RR) in TRPV6 selectivity filter causes closure of the channel gate, presumably due to electrostatic interactions of the positive charge of RR and the electric dipole of the S6 helix. Similarly, we observe that genistein acts both as a blocker of the ion-conducting pathway, by preventing any ion flux through the pore upon its binding, and as a gating modifier that closes the channel gate. This is in stark contrast to another example of TRPV6 blockers, the trivalent cation Gd^{3+} , which occludes the pore by binding at the pore extracellular entry but does not alter the gating state of the channel (Saotome *et al.* 2016; Singh *et al.*, 2017). We have given these and other examples in the text and mentioned that genistein acts as an ion channel blocker and gating-modifier (lines 36, 71, 259-305).

Two genistein molecules are identified per TRPV6 channel in a unique combination of poses.

Given the conformational change these two genistein molecules induce, it is expected that their binding would show cooperativity. However, the function data showed inhibition in an extremely wide, four orders of magnitude concentration range. This observation is puzzling and not fully consistent with the structural model. It is possible calcium imaging, being an indirect measurement, does not faithfully report inhibition of channel activity. If the authors believe this is the case, then they should confirm with patch clamp which they have used in the present and previous studies, and to fit inhibition of channel currents to a Hill (instead of logistic) equation to obtain the Hill slope estimate.

We agree with the Reviewer that calcium imaging is an indirect measurement of channel activity and we did our best to measure the concentration dependence of TRPV6 inhibition by genistein using patch-clamp recordings. The main problem with these measurements turned out to be poor solubility of genistein above 50 μM . We found that at concentrations above 50 μM , our application system was starting to clog up, making measurements in this concentration range unreliable. We therefore measured the concentration dependence with the concentration of genistein up to 50 μM (new Supplementary Fig. 1). Luckily, the IC_{50} value turned out to be lower than 50 μM ($40.7 \pm 2.6 \mu\text{M}$), making the estimation of IC_{50} reasonably reliable. Of course, the estimate of the Hill coefficient value (1.80 ± 0.13) might not be very precise in this concentration range, but we believe it is probably not too much off given the good fit of the data points. The value of the Hill coefficient higher than 1 suggests cooperativity of the two genistein binding sites, in agreement with the Reviewer's suggestion. The new Supplementary Figure 1 and the corresponding text have been added to the manuscript.

Supplementary Fig. 1: Concentration dependence of TRPV6-mediated current inhibition by genistein. **a**, Whole-cell patch-clamp currents recorded from HEK 293 cells expressing hTRPV6 in response to -100 to 70 mV voltage ramp at different concentrations of genistein (labeled, in μM). **b-c**, Concentration dependence of genistein inhibition for individual (**b**) and average (**c**) measurements of TRPV6-mediated current amplitude at -60 mV, normalized to the value in the absence of genistein (0 μM). The curves show fitting of the average normalized current concentration dependence with logistic equation and $\text{IC}_{50} = 40.7 \pm 2.6 \mu\text{M}$ and $n_{\text{Hill}} = 1.80 \pm 0.13$ ($n = 7$ independent experiments).

The significance of metal binding is suggested in the manuscript, but not well established. If the bound metal ions came from the protein preparation stages, as the authors speculated, what might be the physiological metal ions? Would their binding facilitate conformational change to the closed-like one that genistein binding produced (point mutations yielded rather small shifts of the very shallow inhibition curve)? If it is considered physiological, then won't the previously reported closed channels (hTRPV6 in different lipid compositions, rat TRPV6 channels, etc.)

contain the bound metal ions as well? Finding the identity of the bound metal ions, for example using inductively coupled plasma mass spectrometry (ICP-MS) or X-ray energy scan, would be very helpful. Directly testing effects of adding metal ions (would require patch clamp in inside-out configuration) would be even better. In the absence of additional information, it is suggested to de-emphasize this part and make the associated discussion conservative.

We thank the Reviewer for raising this important point. The novel conformation created by the binding of genistein creates a 3-histidine cluster, which has never been observed in other TRPV6 conformations before and according to the coordination geometry of the putative metal ion, likely represents a binding site for zinc (Brodersen et al. 1999, Bebrone et al. 2009, Zhang et al. 2012). We have not observed these ion-like densities at the intracellular pore entrance or other locations in the pore in other TRPV6 structures before, except for the usual coordination of metal ions in the selectivity filter of TRPV6. Although we could use ICP-MS or X-ray energy scan to identify the presence of different ions in our protein sample, we would still not be able to determine the identity of the ions at the proposed tri-histidine sites. The lab has no expertise in inside-out patch-clamp recordings but based on our mutagenesis studies we do not expect strong effects on genistein inhibition. Given the relatively weak effects of mutations in the 3-histidine clusters revealed by our calcium imaging experiments (Figure 4d), we agree that these clusters have only a minor effect on the genistein block and have followed the Reviewer's suggestion to de-emphasize this part and make the associated discussion conservative (lines 38, 73, 218-220).

The term "straight line" should be changed to "curve" (figure legend).

Thank you very much for this observation. We have changed the wording (lines 434, 480).

Jie Zheng

Reviewer #3 (Remarks to the Author):

The paper by Neuberger et al. reported the potential TRPV6 binding sites of genistein, a natural blocker, and the structural changes induced by the binding events. Two binding sites were assigned to the ligand above and below the crossing points of four S6 helices. The cryo-EM structures depicted symmetry breakdown and conformational changes in the specific regions close to the intracellular entry site (S2), which are pretty interesting and make genistein a different type of blockers that might have potential for structure-based drug design should its binding affinity be improved to nM range and be more specific. The paper is of unique interest to the field in both structural studies of ligands of TRP channels and the cryo-EM structure-based drug design. A few points need to be addressed before the paper is ready for publication.

- 1) The two binding sites were assigned without showing details of the ligand densities, especially the matching of the ligand models with the density shape and the bonding structures for stabilizing the ligands in the two binding pockets. As a potential compound of clinical applications, it is especially important to specify if the ligand densities are resolved sufficiently well to define unique orientations and configurations (around totable bonds) of the ligands in the two sites.

pretty high affinity, say $K_d \sim 100$ nM, it might compete for at least one of the assigned binding sites and change the story significantly.

Genistein was purchased from Cayman Chemical (item no. 10005167) as a pure extract from *Styphnolobium japonicum* and dissolved in 100% DMSO. The purity of the compound has been assessed as 98.8% in multiple assays by the manufacturer including HPLC, mass spec., thin-layer chromatography (TLC), UV λ max. analysis and NMR. Given the high resemblance of cryo-EM density and genistein molecule models (see the new Supplementary Fig. 2), we strongly believe that these densities represent genistein molecules with high probability. Since we cannot exclude entirely that impurities of the genistein extract are capable of binding to hTRPV6, we now mention this possibility in the text (lines 197-201): “The cryo-EM density for genistein at site 2 is slightly weaker than at site 1. Since genistein was purified from a natural source, we cannot exclude the possibility that the lower site represents a bound contaminant. However, we believe that such a possibility is highly unlikely, given the resemblance of the density with the molecule of genistein and the high degree ($\sim 99\%$) of the reagent purity, which was verified using multiple methods.”

4). Why was it necessary to incubate genistein with the proteins for 2 hours before freezing? In cell incubation it takes time probably for genistein to cross membranes, but not in purified proteins. Was the result the same if the incubation was brief?

Given the high cost of cryo-EM preparations and data collections, our lab has been using a protocol according to which we incubate the purified target protein with the antagonist for 1-2 hours at room temperature before attempting to make cryo-EM grids. This is based on our earlier studies with various agonists and antagonists of different vanilloid family channels, in which short incubations on ice for a few minutes prior to grid freezing did not result in ligand binding of micromolar-affinity ligands. An important component missing in the purified specimen is the lack of a membrane where a hydrophobic drug can accumulate before finding its way to its binding pocket. Compared to the natural cell membranes, such hydrophobic space for potential drug accumulation is negligible in nanodiscs, which we have used for TRPV6 preparations. Since genistein was found bound to TRPV6 from the very first attempt, we did not consider it necessary to explore shorter applications, especially given the high costs of protein preparation and cryo-EM experimentation.

5) Because of the promiscuity from two binding sites, are the ligand densities sufficiently well resolved to differentiate the rings of the flavone backbone in both sites from the ring structure of CHS, given the abundance of the latter?

The genistein molecule has quite a distinct shape. At site 1, the fit of genistein into density is unambiguous (Supplementary Fig. 2a). CHS is a significantly larger molecule, which in our structures is well resolved in density around the TMD (see panel **a** below). Attempts to fit it into site 1 density inevitably result in clashes (see panel **b** below). Density at site 2 has the size of a genistein molecule but can possibly be fit into this density in two ways, related by 180-degree rotation around the pore axis (Supplementary Fig. 2b). CHS can in principle be fit into the space at site 2 but (1) the density does not match as well as it does for genistein (see panel **c** below), (2) the fit brings the hydrophobic acyl tail of CHS in close contact with the positively charged H582,

and (3) despite the fact that CHS is always present in our TRPV6 samples, the density at site 2 shows up only in the presence of genistein. Therefore, we think that sites 1 and 2 densities are unlikely to represent CHS. Nevertheless, we have now mentioned the possibility for site 2 density to represent a molecule of CHS (lines 201-203).

Density for CHS and attempts to fit CHS into genistein sites 1 and 2. **a**, One of CHS sites around the TRPV6 TMD. **b-c**, Sites 1 (**a**) and 2 (**b**) of genistein viewed intracellularly along the channel pore, with molecules of CHS (yellow) and TRPV6 subunits A and C (green) as well as B and D (pink) shown in sticks and putative density for genistein and the surrounding protein as red and blue mesh, respectively. Red circles in **a** indicate clashes of CHS and the surrounding side chains. Red circle in **b** indicates a close contact of the hydrophobic tail of CHS and H582.

6) Given the membrane permeability of genistein, is it possible that it is enriched in detergent micelles and exerts its effect from the periphery next to the S45 linker and the S6-TRP helix?

We carefully compared annular densities around the TRPV6 TMD next to the S4-S5 linker and S6-TRP helix and we do not see substantial difference between hTRPV6_{Open} and hTRPV6_{GEN}. Given the excellent fit of genistein into densities at sites 1 and 2, we think that the possibility for genistein to exert its effect from the periphery next to the S4-S5 linker and S6-TRP helix is highly unlikely.

7) The term “intracellular entrance” is more suitable for the location of the S2 site, but not so for the S1. It is better to differentiate them in description because of their physical difference.

We have replaced the “intracellular entrance” with the “intracellular half of the TRPV6 pore” (lines 35, 70, 190).

8) The mechanistic description of the morphed conformational changes in a stepwise fashion in the text and in Fig. 5 is highly speculative and might be misleading without other data. It is better to shorten it or put it in discussion.

We agree with the Reviewer and have moved the entire “Mechanism of hTRPV6 inhibition by genistein” section to Discussion.

9) Given the equilibrium in ligand binding to the proteins in solution, can the populations of the different structures be used to deduce energetic scale for genistein binding (likely at S2)? Or can the interaction energy between genistein and the two binding pockets be calculated and differentiate them?

Interestingly, we observe only two populations of particles in the cryo-EM data set: a previously reported open-state conformation and the genistein-inhibited new conformation. It was not possible to separate sub-classes with only one or the other genistein molecule bound and no alternative conformations were observed in multiple classification attempts. Such a task, which would be very interesting to study indeed, might be a more suitable target for an extensive computational study using molecular dynamic simulations. These are, however, outside of the scope and expertise of our lab.

REVIEWER COMMENTS

Reviewer #1 (Remarks to the Author):

The revision satisfactorily addressed my previous concerns. In particular, the Hill slope of 1.8 determined by patch clamping confirms strong cooperativity between the two genistein molecules, as the other reviewer also suggested, and is in agreement with the structural data. No further request.

Reviewer #2 (Remarks to the Author):

The revised manuscript has addressed some of the concerns, but still failed to properly eliminate uncertainties on the main conclusions of the two binding sites. It is therefore premature to publish this manuscript in the current form. Details are in the following.

First, the experiments suggested for differentiating S1 and S2 sites are still needed. With the capacity to do whole-cell recordings, it is easy to do inside-out recordings in the same system. Given the different IC₅₀ and Hill coefficients (0.77 vs. 1.8) from two different assays (Fig 1a-c), it is important to examine the two sites separately because Hill efficient = 1.8 (line 87) does not make sense when occupation one site (either S1 or S2) suffices to block the currents completely.

Second, regard S1 site, the uncertainties come from the following. i) At a 2.66 Å global resolution and the binding sites in the core of the 3D map with even better resolutions (suppl Fig. 2g) the density of S1 ligand should be pretty well resolved such that the density shape matches with that of the compound very well. Supplementary Fig. 3a shows a strong density (purple circle and arrow on the left side) that does not agree with the compound, which should not happen at the current resolution. ii) In 3D space, the 6-member ring and the 10-member ring are expected to be nearly orthogonal to each other (right side). The coplanarity of the two rings in S1 is unexpected and energetically unfavored. iii). The purity (98.8%) of natural product (genistein) is not sufficient to exclude that a slightly different compound (maybe a derivative in trace amounts) may be selected by the TRPV6 channel to generate the induced fit in the S1 site, given the mismatch in i. iv) The binding pocket of S1 is made of all hydrophobic residues (L574, L571 and M570), which is unfavorable for accommodating the polar groups of genistein given the partial negative charges around it (red in the right side). v). The induced conformational changes by S1 occupation suggest a significant free energy change. That estimated from IC₅₀ seems relatively small. A thermodynamic consideration of apo + L < B is needed to account for the observed differences.

(Figures are in the attached doc file)

Third, the uncertainty of S2 site is even more in consideration of the following aspects. I). The shape of the ligand density in S2 site deviates from that of the compound much more significantly (supplementary Fig. 3b) than the S1, given ~2.5 Å from Fig. 2g in the core of the C1 map. Neither the six-member ring nor the 10-member ring matches with the shape of the density. II). The coplanarity of the two rings is not expected, either. III). The S2 occupancy seems to depend on the S1 occupancy, which makes it important to differentiate them experimentally. IV). Given that the hydrophobic tail of CHS is flexible and that the density in the S2 site is much larger in volume than that of a genistein (two blue circles), it is likely that CHS or another molecule, other than genistein, is bound in two opposite orientations (red arrows). These two opposite orientations may be resolved by 3D classification in focused refinement. V) The main hydrophobic body of CHS matches well with the hydrophobic residues (M578), but not the negative partial charges around genistein. VI). Given that the succinate is interacting with H582, it would be interesting to test if H582R would disfavor S2 occupancy by CHS or a similarly charged, amphipathic molecule.

Fourth, the structural details for S1 and S2 sites (supplementary Fig. 3) are pretty important and should be presented in the main text. 3D stereo views are preferred to compare the ligand density shapes and the 3D chemical models.

We are thankful to Reviewer #2 for the suggestions. We have made changes to the manuscript with the details outlined in our responses below.

The revised manuscript has addressed some of the concerns, but still failed to properly eliminate uncertainties on the main conclusions of the two binding sites. It is therefore premature to publish this manuscript in the current form. Details are in the following.

First, the experiments suggested for differentiating S1 and S2 sites are still needed. With the capacity to do whole-cell recordings, it is easy to do inside-out recordings in the same system. Given the different IC₅₀ and Hill coefficients (0.77 vs. 1.8) from two different assays (Fig 1a-c), it is important to examine the two sites separately because Hill efficient = 1.8 (line 87) does not make sense when occupation one site (either S1 or S2) suffices to block the currents completely.

Doing whole-cell recordings is not the same as doing inside-out patch-clamp recordings. While we have expertise in doing the former, we do not have expertise in doing the latter. Besides, there is no difference for differentiating S1 and S2 sites whether the experiments are done in the whole-cell or inside-out modes. The only feasible way of differentiating them is knocking out one of the two sites and testing whether the second site mediates the inhibition. The problem with this approach with respect to genistein is obvious – genistein binding (to both S1 and S2) occurs in the structural region of TRPV6 that undergoes gating-related conformational changes. Both the gate and the ion-conducting channel region are the most sensitive parts of this channel. Every single change we make in this region will inevitably alter TRPV6 gating.

On the issue of IC₅₀ and the Hill coefficient. These values are macroscopic parameters. Correspondingly, the first one is not a measure of microscopic binding constant or binding energy, even though it is related to them. Similarly, the second one is not a measure of ligand binding stoichiometry, even though it is related to it. From our previous structural studies, we know that the TRPV6 conformational ensemble includes three major conformational states (closed, open and inactivated) and in the presence of genistein, it also includes at least one more and novel conformational state represented by the new structure hTRPV6_{GEN}. This means that the kinetic scheme that describes energetics of TRPV6 includes at least 4 different states, but realistically many more states and transitions between them. To properly describe such a complex and non-trivial kinetic system, we would need to make experimental measurements that include determination of a number of independent kinetic parameters that is equal or exceeds the number of transitions. Obviously, our Fura2 measurements, which only provide an estimate of intracellular calcium that results from function of TRPV6 (not necessarily a direct measure of calcium carried through TRPV6) and whole-cell measurements of I-V curves, which are prone to errors associated with uncertainty of estimating the leak current, do not provide the necessary number of measured parameters, nor do they represent direct measurements of any kinetic/thermodynamic parameters (information extracted from them will depend on the kinetic scheme and transitions between its states). Since we do not use experimental approaches that provide estimates of a sufficient number of kinetic parameters, we have been very careful to not overanalyze the data obtained in Fura2 and whole-cell patch-clamp experiments and we were very transparent, conservative, and unbiased in all our interpretations. The reason why we show the results of these experiments is to provide a minimum description of genistein inhibition that can be used later by those who would like to figure out more details about energetics of genistein interaction with TRPV6. From our perspective, the most appropriate experiments that would allow a more detailed view on energetics of inhibition of the

constitutively open channel TRPV6 by genistein are single-channel recordings. This is, however, outside the scope of this manuscript and deserves a separate study in the future.

Overall, it is difficult to assess the relative contribution of sites 1 and 2 to the mechanism of TRPV6 inhibition by genistein. However, given the more stable behavior of genistein at site 1 compared to site 2 (new Fig. 5, new Supplementary Fig. 6) and relatively weak contribution of metal coordination in the vicinity of site 2 to inhibition of calcium influx (Fig. 4d), we hypothesize that site 1 represents the main site of genistein action. The corresponding discussion has been added to the text (lines 309-315).

Second, regard S1 site, the uncertainties come from the following. i) At a 2.66 Å global resolution and the binding sites in the core of the 3D map with even better resolutions (suppl Fig. 2g) the density of S1 ligand should be pretty well resolved such that the density shape matches with that of the compound very well. Supplementary Fig. 3a shows a strong density (purple circle and arrow on the left side) that does not agree with the compound, which should not happen at the current resolution.

Despite very good overall resolution, two major factors prevent genistein from displaying a perfectly fitting density at site S1. First, the site is located at the pseudo-4-fold rotational symmetry axis, where the noise signal is always exaggerated. Second, while we present only one, the most likely pose of genistein fitting in site S1 based on an unbiased refinement algorithm, there are 4 equivalent poses that can represent genistein in 3D reconstruction (related by 180-degree rotations around the pseudo-4-fold rotational symmetry axis of the channel and the axis going through the longest plain of the genistein molecule; see figure below). Attempts to determine a preferable pose through multiple rounds of focused classification and refinement have not succeeded in better resolved density (on top of the problem of the molecule positioning at the rotational symmetry axis, the density is way too small for meaningful focused refinement or classification attempts). The presence of all four poses representing genistein in site S1 is supported by relatively low apparent affinity (relatively high IC₅₀ values) in Fura2 and patch-clamp experiments.

To test our conclusions, we have now run MD simulations starting with genistein placed at site 1 in the four different poses. The results of our MD simulations, illustrated in the new Fig. 5a and Supplementary Fig. 6, show a beautiful match between the MD-predicted and experimental cryo-EM densities, strongly supporting our above conclusion that the experimental density reflects binding of genistein in different poses.

ii) In 3D space, the 6-member ring and the 10-member ring are expected to be nearly orthogonal to each other (right side). The coplanarity of the two rings in S1 is unexpected and energetically unfavored.

The orthogonality might be true in vacuum but the refinement program with the restraints generated in Phenix package suggests that in the site S1 environment, the 6-member ring and the 10-member ring are not orthogonal to each other. However, they are not coplanar either, as clearly illustrated in tilted views of the same image below.

An angle between the 6-member ring and the 10-member ring modelled in our cryo-EM structure is also very similar to the angle predicted by our MD simulations (see new Fig. 5 or Supplementary Fig. 6c as examples).

iii). The purity (98.8%) of natural product (genistein) is not sufficient to exclude that a slightly different compound (maybe a derivative in trace amounts) may be selected by the TRPV6 channel to generate the induced fit in the S1 site, given the mismatch in i. iv) The binding pocket of S1 is made of all hydrophobic residues (L574, L571 and M570), which is unfavorable for accommodating the polar groups of genistein given the partial negative charges around it (red in the right side).

Such concern can be expressed regarding any study involving small molecules (most commercially available purified small molecule ligands from common suppliers such as Sigma, Aldrich, Cayman Chemicals, Tocris etc. are sold as 95-98% purity grades and have been used as such in many previous structural studies). In this case, such a scenario is very unlikely because this would mean that some sort of a very rare impurity of genistein, which happens to look almost exactly like genistein according to the density at site S1, has extremely high affinity towards TRPV6, which is also inconsistent with the relatively high IC₅₀ values. Besides, the fraction of such impurity (< 1% or < 20 μM) would mean that its absolute concentration is smaller than the concentration of protein (> 30 μM), making the occupancy of the sites and correspondingly the fraction of genistein-bound particles much lower than observed experimentally. In addition, the results of our MD stimulations (see new Fig. 5a and Supplementary Fig. 6) clearly demonstrate that binding of genistein to site 1 is very favorable and that the ligand never leaves the site during simulations, being stabilized by two hydrogen bonds between the hydroxyl groups of genistein and the carbonyl oxygens of M570 in subunits A and C.

v). The induced conformational changes by S1 occupation suggest a significant free energy change. That estimated from IC₅₀ seems relatively small. A thermodynamic consideration of apo + L < B is needed to account for the observed differences.

Assuming a much more complicated kinetic scheme than the proposed apo + L < B to describe the overall experimental situation (see our comments above), it is unfeasible to make an appropriate thermodynamic estimate of free energy based on the measurements of IC₅₀ values.

Third, the uncertainty of S2 site is even more in consideration of the following aspects. I). The shape of the ligand density in S2 site deviates from that of the compound much more significantly (supplementary Fig. 3b) than the S1, given ~2.5 Å from Fig. 2g in the core of the C1 map. Neither the six-member ring nor the 10-member ring matches with the shape of the density.

The likely reason for the poorer match of density at site S2 to the molecule of genistein is that the ensemble of genistein positions and poses at this site is more diverse than at site S1. Indeed, our MD simulations demonstrate that multiple poses of genistein at site 2 contribute to the predicted density, which matches amazingly well with the experimental cryo-EM density (see new Fig. 5b and Supplementary Fig. 6). In addition, non-protein densities unaccounted by genistein at the primary position in site 2 are very well predicted by the presence of another genistein molecule at the secondary location in site 2, in the close proximity to the primary position. This second genistein molecule at the secondary position in site 2 is even more mobile than the one at the primary position and we therefore do not focus on it as a separate site for simplicity (see Supplementary Fig. 6).

II). The coplanarity of the two rings is not expected, either.

Similar to site S1 (see the illustration above), the coplanarity is not observed at the site S2 either (see, for example, new Fig. 5b and Supplementary Fig. 6c).

III). The S2 occupancy seems to depend on the S1 occupancy, which makes it important to differentiate them experimentally.

The only feasible way of differentiating them is knocking out one of the two sites and testing whether the second site mediates the inhibition. The problem of this approach is obvious – genistein binding occurs in the structural region of TRPV6 which undergoes gating-related conformational changes. According to our experience, every single mutation in this region will inevitably alter TRPV6 gating. Correspondingly, the effects on gating and on genistein binding will be impossible to separate.

IV). Given that the hydrophobic tail of CHS is flexible and that the density in the S2 site is much larger in volume than that of a genistein (two blue circles), it is likely that CHS or another molecule, other than genistein, is bound in two opposite orientations (red arrows). These two opposite orientations may be resolved by 3D classification in focused refinement.

We agree that it is possible for CHS in a folded non-linear conformation to be jammed in and fitted into site S2. Given this possibility, we have already corrected our interpretations in the previous resubmission, clearly stating that it is possible that the density at site S2 can represent not only genistein but also other molecules, like CHS. We have also made numerous attempts to resolve the two orientations proposed by the Reviewer using focused classification and refinement but without success. In addition, for this resubmission, we have also run MD simulations by incorporating CHS in different poses at site 2 (new Supplementary Fig. 7). As clearly illustrated by the results of our MD simulations, the MD-predicted density for CHS is much more extended and does not resemble the experimental density. Given the excellent match of the experimental density with the MD-predicted density for genistein (see above), the experimentally predicted density in site 2 unlikely represents CHS.

V) The main hydrophobic body of CHS matches well with the hydrophobic residues (M578), but not the negative partial charges around genistein.

Genistein is a neutral molecule that carries no net charge, while CHS carries a -1 charge at its carboxyl group. From this perspective, fitting CHS into site S2 binding pocket is much less energetically favorable than fitting genistein. In addition, as we mentioned in our previous response, the site S2 is unlikely to represent CHS for the following reasons: (1) the density does not match CHS as well as it matches genistein, (2) the fit brings the hydrophobic acyl tail of CHS in close contact with the positively charged H582 (if the CHS molecule is flipped as illustrated by Reviewer #2, H582 from the diagonally opposed subunit will make the exactly same clash), and (3) despite the fact that CHS is always present in our TRPV6 samples, the density at site 2 shows up only in the presence of genistein (we have never observed anything like this in any of our previous preparations – all of which were done in exactly the same way, i.e. in the presence of CHS). In addition, our new MD experiments provide strong support to the conclusion that site 2 accommodates genistein much better than CHS. Indeed, placing CHS at site 2 results in MD-predicted density (see new Supplementary Fig. 7) that matches the experimental density much more poorly than the MD-predicted density for genistein (see new Fig. 5b and Supplementary Fig. 6).

VI). Given that the succinate is interacting with H582, it would be interesting to test if H582R would disfavor S2 occupancy by CHS or a similarly charged, amphipathic molecule.

As we have already mentioned above, the problem of the mutagenesis approach is obvious – genistein binding occurs in the structural region of TRPV6, which undergoes gating-related conformational changes. Every single change we make in this region will inevitably alter TRPV6 gating. Correspondingly, the effects on gating and on genistein binding will be impossible to separate.

Fourth, the structural details for S1 and S2 sites (supplementary Fig. 3) are pretty important and should be presented in the main text. 3D stereo views are preferred to compare the ligand density shapes and the 3D chemical models.

We have now introduced an additional figure (new Fig. 5) that illustrates the MD-predicted density along with the experimental cryo-EM density in the vicinity of sites 1 and 2. We feel this is sufficient representation for the main figures, given that the article files for Nature Communication papers include all supplementary materials items that directly follow the main body of the paper. We are hesitant to introduce a stereo view figure as this form of presentation is not anymore popular as it used to be (probably because structural information is used by a broader range of scientists but most of them do not know how to look at stereo views).

REVIEWER COMMENTS

Reviewer #2 (Remarks to the Author):

The revised version of the manuscript by Neuberger, et al. has added data from all-atom MD simulations for both sites 1 and 2. Some of the concerns raised last time were resolved, such as that the compound poses show non-coplanarity of the 6-member and 10-member rings as I expected. But the data are still not sufficient to endow high level of certainty to the main conclusion of two different binding sites of genistein in the TRPV6_{GEN} structure, which does not support the publication of this manuscript.

The experimental difficulty expressed by the authors in doing inside-out patch recordings to access the sites 1 and 2 directly from the intracellular side sounds overstated, and should not be a road-block to experimental studies. From what the reviewer #1 has commented, both his/her lab and mine have expertise in doing inside-out recordings and we know the level of difficulty in whole-cell recordings is similar to or even higher than that of the inside-out. Also, there are excellent experts in patch clamp on the authors' campus, who can provide good technical guidance or support. The concerns on the differences in responses from current recordings and Ca²⁺ signals remain unresolved, main because the changes in recording currents and in Ca-signals were expected to directly connect to changes in total conductance (G) in cell membranes.

Even though there is little doubt that the inhibitor bound at site 1 is structurally very close to genistein, there is a striking density feature unaccounted for (pose 1 in the above Fig., pointed

by red arrows), suggesting an additional group (like -OH, -CH₃, -NH₂, etc.) that might be attached to the assumed chemical model of genistein. At the reported resolution of 2.6 Å, such a strong density suggests that the real compound in site 1 is likely a derivative of genistein. Given that only a fraction (~8.4% from suppl. table) of the total particles were used in 3D reconstruction of the ligand-bound structure, it was risky to think that 20 micromolar contaminants out of the total 2 mM were not enough to lead to sufficient molecules with occupied site 1 in the sample. Verifying the true chemical nature of the compound bound in site 1 might lead to something of higher potency, and may be critical for future structure-based drug design, should the current study of genistein be suitable for such a development.

The second binding site still bears ample uncertainty. The density at site 2 requires the ligand occupancy in site 1. The new MD data in supplementary Fig. 7d suggested much stable conformations for CHS, starting with all four poses, equivalent to or even slightly better than the proposed genistein in site 1 in stability (lower displacements; supplementary Fig. 7d vs. suppl. Fig. 6b), and better in stability than the proposed primary and secondary sites of genistein in site 2 (supplementary Figs. 6c,d), especially the latter. This is probably due to the strong electrostatic interactions between R589 and succinate. The ensembled density for CHS accounts for the core EM density of the ligand in site 2 well, despite more density due to molecular motions, similar to what was seen in the ensembled density of genistein in site 1 (Fig 5a). The fact that in MD simulation, it is necessary to introduce two molecules to account for the ligand EM density in site 2, and the poor agreement of the secondary site with the density (supplementary Fig. 6a) further erode the conclusion that site 2 is not CHS or other molecules. In addition, importance of R589 and other parts to form the site 2 does not support the argument that because CHS does not bind to the apo-TRPV6 channel, it should not be present in site 2. Resolving this concern sufficiently is thus critical to the main conclusion of this manuscript and this whole study. More experimental evidence is required. For example, is it feasible to remove CHS completely from the specimens, or mutate R589 to test?

A couple of minor concerns:

There is an additional claim of genistein as a gating modifier besides pore blocker. These two seem to point to the same effect of pore-blocking and can not be differentiated from the data presented.

Also, the use of the term oncochannel was apparently intended to suggest that TRPV6 acts like an oncogene, instead of more likely a secondary effector after cancer development. It thus appears an over-exaggeration.

The reported clash scores are somewhat higher than expected for the achieved high-resolutions (2.6 and 2.7). Given that H-bonds are important for MD calculations, it probably will be helpful to use ISODE to further refine the structural model of TRPV6_{GEN} and decrease clashes.

We are thankful to Reviewer #2 for the additional suggestions. We have made changes to the manuscript with the details outlined in our responses below.

The revised version of the manuscript by Neuberger, et al. has added data from all-atom MD simulations for both sites 1 and 2. Some of the concerns raised last time were resolved, such as that the compound poses show non-coplanarity of the 6-member and 10-member rings as I expected. But the data are still not sufficient to endow high level of certainty to the main conclusion of two different binding sites of genistein in the TRPV6GEN structure, which does not support the publication of this manuscript.

Our state-of-the-art MD simulations, which were elaborate, time-consuming, and rigorous, have been added to our previous submission to specifically address the concerns of Reviewer #2. The main conclusions of our MD simulations experiments are that the experimental cryo-EM density is highly consistent with genistein molecules present at sites 1 and 2 and inconsistent with any presence of CHS. The results of our MD simulations therefore strongly and unambiguously support two binding sites of genistein.

The experimental difficulty expressed by the authors in doing inside-out patch recordings to access the sites 1 and 2 directly from the intracellular side sounds overstated, and should not be a road-block to experimental studies. From what the reviewer #1 has commented, both his/her lab and mine have expertise in doing inside-out recordings and we know the level of difficulty in whole-cell recordings is similar to or even higher than that of the inside-out. Also, there are excellent experts in patch clamp on the authors' campus, who can provide good technical guidance or support. The concerns on the differences in responses from current recordings and Ca²⁺ signals remain unresolved, main because the changes in recording currents and in Ca-signals were expected to directly connect to changes in total conductance (G) in cell membranes.

As we had explained during our previous submission, inside-out patch-clamp recordings will not help to resolve the main remaining concerns of Reviewer #2. Given that Reviewer #1, who originally suggested to do these experiments, has agreed with us, we prefer not to perform such experiments ourselves and leave this opportunity to others who are interested.

Even though there is little doubt that the inhibitor bound at site 1 is structurally very close to genistein, there is a striking density feature unaccounted for (pose 1 in the above Fig., pointed by red arrows), suggesting an additional group (like -OH, -CH₃, -NH₂, etc.) that might be attached to the assumed chemical model of genistein. At the reported resolution of 2.6 Å, such a strong density suggests that the real compound in site 1 is likely a derivative of genistein. Given that only a fraction (~8.4% from suppl. table) of the total particles were used in 3D reconstruction of the ligand-bound structure, it was risky to think that 20 micromolar contaminants out of the total 2 mM were not enough to lead to sufficient molecules with occupied site 1 in the sample. Verifying the true chemical nature of the compound bound in site

1 might lead to something of higher potency, and may be critical for future structure-based drug design, should the current study of genistein be suitable for such a development.

The bump of density that Reviewer #2 refers to can easily be noise (the location of the site is at the pseudo-4-fold axis of symmetry where noise is exaggerated – as we have already mentioned and explained in our previous submissions) or a part of the genistein molecule. In fact, upon a careful inspection of the MD simulations-predicted density in Fig. 5, it becomes obvious that such bump is rather expected to be present. We would therefore refrain from overinterpretation of our data by suggesting that there could be other molecules at site 1 rather than genistein because the chances of this (as we discussed in detail in the previous submission) are slim. Besides, by inspecting the recently deposited high-resolution cryo-EM structures of membrane proteins in complex with small-molecules, the probability of finding a perfect cryo-EM density with exact shape of the molecule is close to zero. In our case, however, the shape of the genistein cryo-EM density, especially at site 1, is quite frankly as good as it can get.

Also, it should be noted that we used only a fraction of particles for the final map in order to achieve the highest quality reconstruction. Every cryo-EM data processing pipeline starts with hundreds of thousands to several millions of template-picked particles – the vast majority of which is ‘junk’ which needs to be cleaned up in iterative rounds of 2D and 3D classifications. We could have easily included more particles in the final reconstruction at the price of a lower quality/resolution map. In fact, in the process of curation and cleanup, we processed several dozens of maps that included magnitudes of more particles, but which also resulted in lower quality/resolution reconstructions. At no point during our rigorous data processing did we notice any alternative central density shapes than the ones reported in the final map.

The second binding site still bears ample uncertainty. The density at site 2 requires the ligand occupancy in site 1. The new MD data in supplementary Fig. 7d suggested much stable conformations for CHS, starting with all four poses, equivalent to or even slightly better than the proposed genistein in site 1 in stability (lower displacements; supplementary Fig. 7d vs. suppl. Fig. 6b), and better in stability than the proposed primary and secondary sites of genistein in site 2 (supplementary Figs. 6c,d), especially the latter. This is probably due to the strong electrostatic interactions between R589 and succinate. The ensembled density for CHS accounts for the core EM density of the ligand in site 2 well, despite more density due to molecular motions, similar to what was seen in the ensembled density of genistein in site 1 (Fig 5a). The fact that in MD simulation, it is necessary to introduce two molecules to account for the ligand EM density in site 2, and the poor agreement of the secondary site with the density (supplementary Fig. 6a) further erode the conclusion that site 2 is not CHS or other molecules. In addition, importance of R589 and other parts to form the site 2 does not support the argument that because CHS does not bind to the apo-TRPV6 channel, it should not be present in site 2. Resolving this concern sufficiently is thus critical to the main conclusion of this manuscript and this whole study. More experimental evidence is required. For example, is it feasible to remove CHS completely from the specimens, or mutate R589 to test?

We cannot possibly remove CHS from the preparation of our protein because it becomes heavily unstable: TRPV6 requires substitutes of cholesterol and we clearly visualize the corresponding densities in every high-resolution structure of TRPV6, at three conservative sites per subunit. We cannot mutate R589 either because this residue forms a crucial open-state stabilizing interaction with Q473 (see our previously published work: McGoldrick et al., Nature 2018; doi:10.1038/nature25182). Disturbing this interaction will prevent channel opening (and functional characterization accordingly) and will therefore be prohibitive for genistein binding. To address the remaining concern of Reviewer #2, we have now emphasized in the text even stronger that while (1) we are 100% certain that site 1 is occupied by genistein and (2) our MD simulations suggest that site 2 is also occupied by genistein, we fully acknowledge the possibility that site 2 can be occupied by something else, for example, by a molecule of CHS (lines 200-207, 279-281). We have also removed the mentioning of the two genistein binding sites from the Abstract and replaced “genistein binds to two sites at the intracellular half of the TRPV6 pore” with “genistein binds in the intracellular half of the TRPV6 pore” (line 33). A similar change has been made in the Introduction (line 70).

A couple of minor concerns:

There is an additional claim of genistein as a gating modifier besides pore blocker. These two seem to point to the same effect of pore-blocking and can not be differentiated from the data presented.

We agree. This is why we simply phenomenologically acknowledge the fact that genistein sits in the pore and physically occludes it as a pore blocker, but its presence also converts the region of the TRPV6 gate to a conformation different from the open state conformation.

Also, the use of the term oncochannel was apparently intended to suggest that TRPV6 acts like an oncogene, instead of more likely a secondary effector after cancer development. It thus appears an over-exaggeration.

TRPV6 has been classified as an oncochannel (doi: 10.1016/j.ceca.2013.01.001; doi: 10.7150/jca.31640) and its gene as an oncogene (doi: 10.1113/jphysiol.2011.225862; doi: 10.1007/978-94-007-0265-3_49; doi: 10.1007/978-94-007-0265-3_48). Since we are not specialists in cancer, we will stick to terminology proposed by the specialists.

The reported clash scores are somewhat higher than expected for the achieved high-resolutions (2.6 and 2.7). Given that H-bonds are important for MD calculations, it probably will be helpful to use ISODE to further refine the structural model of TRPV6_{GEN} and decrease clashes.

We used phenix_real_space_refine to re-refine our structures and reduced clash scores from 7.28 to 5.10 for TRPV6_{GEN} and from 8.43 to 6.76 for TRPV6_{Open}. The Supplementary Table has been updated accordingly.